# A new genus of horse from Pleistocene North America

Peter D Heintzman[1,2]*, Grant D Zazula[3], Ross DE MacPhee[4], Eric Scott[5,6], James A Cahill[1], Brianna K McHorse[7], Joshua D Kapp[1], Mathias Stiller[1,8], Matthew J Wooller[9,10], Ludovic Orlando[11,12], John Southon[13], Duane G Froese[14], Beth Shapiro[1,15]*

[1]Department of Ecology and Evolutionary Biology, University of California, Santa Cruz, Santa Cruz, United States; [2]Tromsø University Museum, UiT - The Arctic University of Norway, Tromsø, Norway; [3]Yukon Palaeontology Program, Government of Yukon, Whitehorse, Canada; [4]Department of Mammalogy, Division of Vertebrate Zoology, American Museum of Natural History, New York, United States; [5]Cogstone Resource Management, Incorporated, Riverside, United States; [6]California State University San Bernardino, San Bernardino, United States; [7]Department of Organismal and Evolutionary Biology, Harvard University, Cambridge, United States; [8]Department of Translational Skin Cancer Research, German Consortium for Translational Cancer Research, Essen, Germany; [9]College of Fisheries and Ocean Sciences, University of Alaska Fairbanks, Fairbanks, United States; [10]Alaska Stable Isotope Facility, Water and Environmental Research Center, University of Alaska Fairbanks, Fairbanks, United States; [11]Centre for GeoGenetics, Natural History Museum of Denmark, København K, Denmark; [12]Université Paul Sabatier, Université de Toulouse, Toulouse, France; [13]Keck-CCAMS Group, Earth System Science Department, University of California, Irvine, Irvine, United States; [14]Department of Earth and Atmospheric Sciences, University of Alberta, Edmonton, Canada; [15]UCSC Genomics Institute, University of California, Santa Cruz, Santa Cruz, United States

*For correspondence:
peteheintzman@gmail.com (PDH);
bashapir@ucsc.edu (BS)

Competing interests: The authors declare that no competing interests exist.

**Abstract** The extinct 'New World stilt-legged', or NWSL, equids constitute a perplexing group of Pleistocene horses endemic to North America. Their slender distal limb bones resemble those of Asiatic asses, such as the Persian onager. Previous palaeogenetic studies, however, have suggested a closer relationship to caballine horses than to Asiatic asses. Here, we report complete mitochondrial and partial nuclear genomes from NWSL equids from across their geographic range. Although multiple NWSL equid species have been named, our palaeogenomic and morphometric analyses support the idea that there was only a single species of middle to late Pleistocene NWSL equid, and demonstrate that it falls outside of crown group *Equus*. We therefore propose a new genus, *Haringtonhippus*, for the sole species *H. francisci*. Our combined genomic and phenomic approach to resolving the systematics of extinct megafauna will allow for an improved understanding of the full extent of the terminal Pleistocene extinction event.
DOI: https://doi.org/10.7554/eLife.29944.001

## Introduction

The family that includes modern horses, asses, and zebras, the Equidae, is a classic model of macro-evolution. The excellent fossil record of this family clearly documents its ~55 million year evolution

**eLife digest** The horse family – which also includes zebras, donkeys and asses – is often featured on the pages of textbooks about evolution. All living horses belong to a group, or genus, called *Equus.* The fossil record shows how the ancestors of these animals evolved from dog-sized, three-toed browsers to larger, one-toed grazers. This process took around 55 million years, and many members of the horse family tree went extinct along the way.

Nevertheless, the details of the horse family tree over the past 2.5 million years remain poorly understood. In North America, horses from this period – which is referred to as the Pleistocene – have been classed into two major groups: stout-legged horses and stilt-legged horses. Both groups became extinct near the end of the Pleistocene in North America, and it was not clear how they relate to one another. Based on their anatomy, many scientists suggested that stilt-legged horses were most closely related to modern-day asses living in Asia. Yet, other studies using ancient DNA placed the stilt-legged horses closer to the stout-legged horses.

Heintzman et al. set out to resolve where the stilt-legged horses sit within the horse family tree by examining more ancient DNA than the previous studies. The analyses showed that the stilt-legged horses were much more distinct than previously thought. In fact, contrary to all previous findings, these animals actually belonged outside of the genus *Equus*. Heintzman et al. named the new genus for the stilt-legged horses *Haringtonhippus*, and showed that all stilt-legged horses belonged to a single species within this genus, *Haringtonhippus francisci.*

Together these new findings provide a benchmark for reclassifying problematic fossil groups across the tree of life. A similar approach could be used to resolve the relationships in other problematic groups of Pleistocene animals, such as mammoths and bison. This would give scientists a more nuanced understanding of evolution and extinction during this period.
DOI: https://doi.org/10.7554/eLife.29944.002

from dog-sized hyracotheres through many intermediate forms and extinct offshoots to present-day *Equus*, which comprises all living equid species (*MacFadden, 1992*). The downside of this excellent fossil record is that many dubious fossil equid taxa have been erected, a problem especially acute within Pleistocene *Equus* of North America (*Macdonald et al., 1992*). While numerous species are described from the fossil record, molecular data suggest that most belonged to, or were closely related to, a single, highly variable stout-legged caballine species that includes the domestic horse, *E. caballus* (*Weinstock et al., 2005*). The enigmatic and extinct 'New World stilt-legged' (NWSL) forms, however, exhibit a perplexing mix of morphological characters, including slender, stilt-like distal limb bones with narrow hooves reminiscent of extant Eurasian hemionines, the Asiatic wild asses (*E. hemionus*, *E. kiang*) (*Eisenmann, 1992*; *Eisenmann et al., 2008*; *Harington and Clulow, 1973*; *Lundelius and Stevens, 1970*; *Scott, 2004*), and dentitions that have been interpreted as more consistent with either caballine horses (*Lundelius and Stevens, 1970*) or hemionines (*MacFadden, 1992*).

On the basis of their slender distal limb bones, the NWSL equids have traditionally been considered as allied to hemionines (e.g. *Eisenmann et al., 2008*; *Guthrie, 2003*; *Scott, 2004*; *Skinner and Hibbard, 1972*). Palaeogenetic analyses based on mitochondrial DNA (mtDNA) have, however, consistently placed NWSL equids closer to caballine horses (*Barrón-Ortiz et al., 2017*; *Der Sarkissian et al., 2015*; *Orlando et al., 2008*, *2009*; *Vilstrup et al., 2013*; *Weinstock et al., 2005*). The current mtDNA-based phylogenetic model therefore suggests that the stilt-legged morphology arose independently in the New and Old Worlds (*Weinstock et al., 2005*) and may represent convergent adaptations to arid climates and habitats (*Eisenmann, 1985*). However, these models have been based on two questionable sources. The first is based on 15 short control region sequences (<1000 base pairs, bp; *Barrón-Ortiz et al., 2017*; *Weinstock et al., 2005*), a data type that can be unreliable for resolving the placement of major equid groups (*Der Sarkissian et al., 2015*; *Orlando et al., 2009*). The second consist of two mitochondrial genome sequences (*Vilstrup et al., 2013*) that are either incomplete or otherwise problematic (see Results). Given continuing uncertainty regarding the phylogenetic placement of NWSL equids—which impedes our understanding of Pleistocene equid evolution in general—we therefore sought to resolve their position using multiple mitochondrial and

partial nuclear genomes from specimens representing as many parts of late Pleistocene North America as possible.

The earliest recognized NWSL equid fossils date to the late Pliocene/early Pleistocene (~2–3 million years ago, Ma) of New Mexico (*Azzaroli and Voorhies, 1993*; *Eisenmann, 2003*; *Eisenmann et al., 2008*). Middle and late Pleistocene forms tended to be smaller in stature than their early Pleistocene kin, and ranged across southern and extreme northwestern North America (i.e. eastern Beringia, which includes Alaska, USA and Yukon Territory, Canada). NWSL equids have been assigned to several named species, such as *E. conversidens* Owen 1869, *E. tau* Owen 1869, *E. francisci Hay (1915)*, *E. calobatus* Troxell 1915, and *E. (Asinus)* cf. *kiang*, but there is considerable confusion and disagreement regarding their taxonomy. Consequently, some researchers have chosen to refer to them collectively as *Equus* (*Hemionus*) spp. (*Guthrie, 2003*; *Scott, 2004*), or avoid a formal taxonomic designation altogether (*Der Sarkissian et al., 2015*; *Vilstrup et al., 2013*; *Weinstock et al., 2005*). Using our phylogenetic framework and comparisons between specimens identified by palaeogenomics and/or morphology, we attempted to determine the taxonomy of middle-late Pleistocene NWSL equids.

Radiocarbon ($^{14}$C) dates from Gypsum Cave, Nevada, confirm that NWSL equids persisted in areas south of the continental ice sheets during the last glacial maximum (LGM; ~26–19 thousand years before present (ka BP); *Clark et al., 2009*) until near the terminal Pleistocene, ~13 thousand radiocarbon years before present ($^{14}$C ka BP) (*Weinstock et al., 2005*), soon after which they became extinct, along with their caballine counterparts and most other coeval species of megafauna (*Koch and Barnosky, 2006*). This contrasts with dates from unglaciated eastern Beringia, where NWSL equids were seemingly extirpated locally during a relatively mild interstadial interval centered on ~31 $^{14}$C ka BP (*Guthrie, 2003*), thus prior to the LGM (*Clark et al., 2009*), final loss of caballine horses (*Guthrie, 2003*; *2006*), and arrival of humans in the region (*Guthrie, 2006*). The apparently discrepant extirpation chronology between NWSL equids south and north of the continental ice sheets implies that their populations responded variably to demographic pressures in different parts of their range, which is consistent with results from some other megafauna (*Guthrie, 2006*; *Zazula et al., 2014*; *Zazula et al., 2017*). To further test this extinction chronology, we generated new radiocarbon dates from eastern Beringian NWSL equids.

We analyzed 26 full mitochondrial genomes and 17 partial nuclear genomes from late Pleistocene NWSL equids, which revealed that individuals from both eastern Beringia and southern North America form a single well-supported clade that falls outside the diversity of *Equus* and diverged from the lineage leading to *Equus* during the latest Miocene or early Pliocene. This novel and robust phylogenetic placement warrants the recognition of NWSL equids as a distinct genus, which we here name *Haringtonhippus*. After reviewing potential species names and conducting morphometric and anatomical comparisons, we determined that, based on the earliest-described specimen bearing diagnosable features, *francisci* Hay is the most well-supported species name. We therefore refer the analyzed NWSL equid specimens to *H. francisci*. New radiocarbon dates revealed that *H. francisci* was extirpated in eastern Beringia ~14 $^{14}$C ka BP. In light of our analyses, we review the Plio-Pleistocene evolutionary history of equids, and the implications for the systematics of equids and other Pleistocene megafauna.

## Results

### Phylogeny of North American late Pleistocene and extant equids

We reconstructed whole mitochondrial genomes from 26 NWSL equids and four New World caballine *Equus* (two *E. lambei*, two *E.* cf. *scotti*). Using these and mitochondrial genomes of representatives from all extant and several late Pleistocene equids, we estimated a mitochondrial phylogeny, using a variety of outgroups (Appendix 1, *Appendix 2—tables 1–2*, and *Supplementary file 1*). The resulting phylogeny is mostly consistent with previous studies (*Der Sarkissian et al., 2015*; *Vilstrup et al., 2013*), including confirmation of NWSL equid monophyly (*Weinstock et al., 2005*). However, we recover a strongly supported placement of the NWSL equid clade outside of crown group diversity (*Equus*), but closer to *Equus* than to *Hippidion* (*Figure 1*, *Figure 1—figure supplement 1a*, *Figure 1—source data 1*, and *Appendix 2—tables 1–2*). In contrast, previous palaeogenetic studies placed the NWSL equids within crown group *Equus*, closer to caballine horses than to

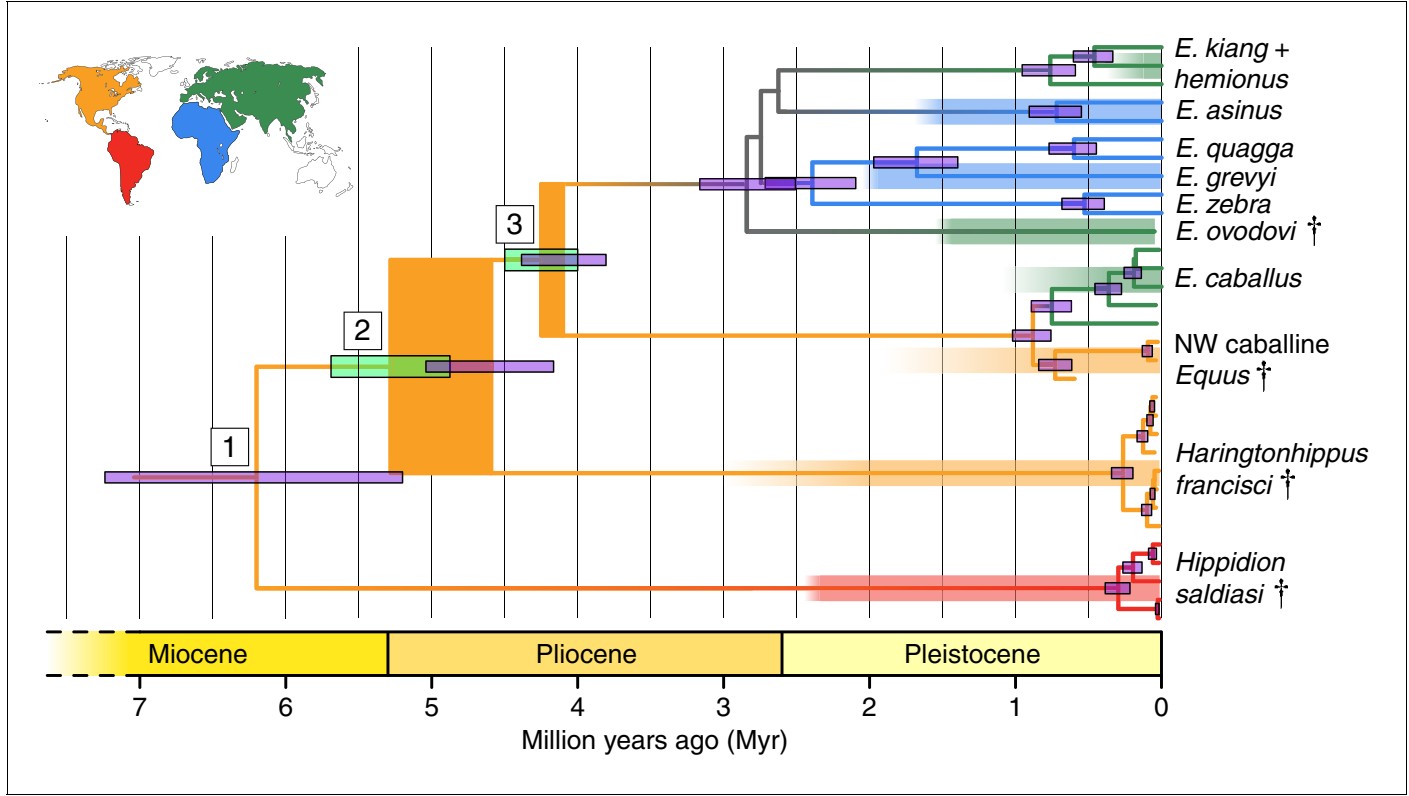

**Figure 1.** Phylogeny of extant and middle-late Pleistocene equids, as inferred from the Bayesian analysis of full mitochondrial genomes. Purple node-bars illustrate the 95% highest posterior density of node heights and are shown for nodes with >0.99 posterior probability support. The range of divergence estimates derived from our nuclear genomic analyses is shown by the thicker, lime green node-bars ([*Orlando et al., 2013*]; this study). Nodes highlighted in the main text are labeled with boxed numbers. All analyses were calibrated using as prior information a caballine/non-caballine *Equus* divergence estimate of 4.0–4.5 Ma (*Orlando et al., 2013*) at node 3, and, in the mitochondrial analyses, the known ages of included ancient specimens. The thicknesses of nodes 2 and 3 represent the range between the median nuclear and mitochondrial genomic divergence estimates. Branches are coloured based on species provenance and the most parsimonious biogeographic scenario given the data, with gray indicating ambiguity. Fossil record occurrences for major represented groups (including South American *Hippidion*, New World stilt-legged equids, and Old World Sussemiones) are represented by the geographically coloured bars, with fade indicating uncertainty in the first appearance datum (after (*Eisenmann et al., 2008*; *Forsten, 1992*; *O'Dea et al., 2016*; *Orlando et al., 2013*) and references therein). The Asiatic ass species (*E. kiang*, *E. hemionus*) are not reciprocally monophyletic based on the analyzed mitochondrial genomes, and so the Asiatic ass clade is shown as '*E. kiang + hemionus*'. Daggers denote extinct taxa. NW: New World.

DOI: https://doi.org/10.7554/eLife.29944.003

The following source data and figure supplements are available for figure 1:

**Source data 1.** Bayesian time tree analysis results, with support and estimated divergence times for major nodes, and the tMRCAs for *Haringtonhippus*, *E. asinus*, and *E. quagga* summarized.
DOI: https://doi.org/10.7554/eLife.29944.007

**Source data 2.** Statistics from the phylogenetic inference analyses of nuclear genomes using all four approaches.
DOI: https://doi.org/10.7554/eLife.29944.008

**Source data 3.** Summary of nuclear genome data from all 17 NWSL equids pooled together and analyzed using approach four.
DOI: https://doi.org/10.7554/eLife.29944.009

**Figure supplement 1.** An example maximum likelihood (ML) phylogeny of equid mitochondrial genomes.
DOI: https://doi.org/10.7554/eLife.29944.004

**Figure supplement 2.** A comparison of relative private transversion frequencies between the nuclear genomes of a horse, donkey, and 17 NWSL equids.
DOI: https://doi.org/10.7554/eLife.29944.005

**Figure supplement 3.** Calculation of divergence date estimates from nuclear genome data.
DOI: https://doi.org/10.7554/eLife.29944.006

non-caballine asses and zebras (*Barrón-Ortiz et al., 2017*; *Der Sarkissian et al., 2015*; *Orlando et al., 2008*, *2009*; *Vilstrup et al., 2013*; *Weinstock et al., 2005*). To explore possible causes for this discrepancy, we reconstructed mitochondrial genomes from previously sequenced NWSL equid specimens and used a maximum likelihood evolutionary placement algorithm (*Berger et al., 2011*) to place these published sequences in our phylogeny *a posteriori*. These analyses suggested that previous results were likely due to a combination of outgroup choice and the use of short, incomplete, or problematic mtDNA sequences (Appendix 2 and *Appendix 2—table 3*).

To confirm the mtDNA result that NWSL equids fall outside of crown group equid diversity, we sequenced and compared partial nuclear genomes from 17 NWSL equids to a caballine (horse) and a non-caballine (donkey) reference genome. After controlling for reference genome and ancient DNA fragment length artifacts (Appendices 1–2), we examined differences in relative private transversion frequency between these genomes (*Appendix 1—figure 1*). We found that the relative private transversion frequency for NWSL equids was ~1.4–1.5 times greater than that for horse or donkey (Appendix 2, *Figure 1—source data 3*, *Figure 1—figure supplement 2*, and *Figure 1—source data 2*). This result supports the placement of NWSL equids as sister to the horse-donkey clade (*Figure 1—figure supplement 3*), the latter of which is representative of living *Equus* diversity (e.g. [*Der Sarkissian et al., 2015*; *Jónsson et al., 2014*]) and is therefore congruent with the mitochondrial genomic analyses.

## Divergence times of *Hippidion*, NWSL equids, and *Equus*

We estimated the divergence times between the lineages leading to *Hippidion*, the NWSL equids, and *Equus*. We first applied a Bayesian time-tree approach to the whole mitochondrial genome data. This gave divergence estimates for the *Hippidion*-NWSL/*Equus* split (node 1) at 5.15–7.66 Ma, consistent with (*Der Sarkissian et al., 2015*), the NWSL-*Equus* split (node 2) at 4.09–5.13 Ma, and the caballine/non-caballine *Equus* split (node 3) at 3.77–4.40 Ma (*Figure 1* and *Figure 1—source data 1*). These estimates suggest that the NWSL-*Equus* mitochondrial split occurred only ~500 thousand years (ka) prior to the caballine/non-caballine *Equus* split. We then estimated the NWSL-*Equus* divergence time using relative private transversion frequency ratios between the nuclear genomes, assuming a caballine/non-caballine *Equus* divergence estimate of 4–4.5 Ma (*Orlando et al., 2013*) and a genome-wide strict molecular clock (following [*Heintzman et al., 2015*]). This analysis yielded a divergence estimate of 4.87–5.69 Ma (*Figure 1—figure supplement 3*), which overlaps with that obtained from the relaxed clock analysis of whole mitochondrial genome data (*Figure 1*). These analyses suggest that the NWSL equid and *Equus* clades diverged during the latest Miocene or early Pliocene (4.1–5.7 Ma; late Hemphillian or earliest Blancan).

## Systematic palaeontology

The genus *Equus* (*Linnaeus, 1758*) was named to include three living equid groups – horses (*E. caballus*), donkeys (*E. asinus*), and zebras (*E. zebra*) – whose diversity comprises all extant, or crown group, equids. Previous palaeontological and palaeogenetic studies have uniformly placed NWSL equids within the diversity of extant equids and therefore this genus (*Barrón-Ortiz et al., 2017*; *Bennett, 1980*; *Der Sarkissian et al., 2015*; *Harington and Clulow, 1973*; *Orlando et al., 2008*; *2009*; *Scott, 2004*; *Vilstrup et al., 2013*; *Weinstock et al., 2005*). This, however, conflicts with the phylogenetic signal provided by palaeogenomic data, which strongly suggest that NWSL equids fall outside the confines of the equid crown group (*Equus*). Nor is there any morphological or genetic evidence warranting the assignment of NWSL equids to an existing extinct taxon such as *Hippidion*. We therefore erect a new genus for NWSL equids, *Haringtonhippus*, as defined and delimited below:

Order: Perissodactyla, Owen 1848
Family: Equidae, Linnaeus 1758
Subfamily: Equinae, Steinmann & Döderlein 1890
Tribe: Equini, Gray 1821
Genus: *Haringtonhippus*, gen. nov. urn:lsid:zoobank.org:act:35D901A7-65F8-4615-9E13-52A263412F67
*Type species. Haringtonhippus francisci* Hay 1915.

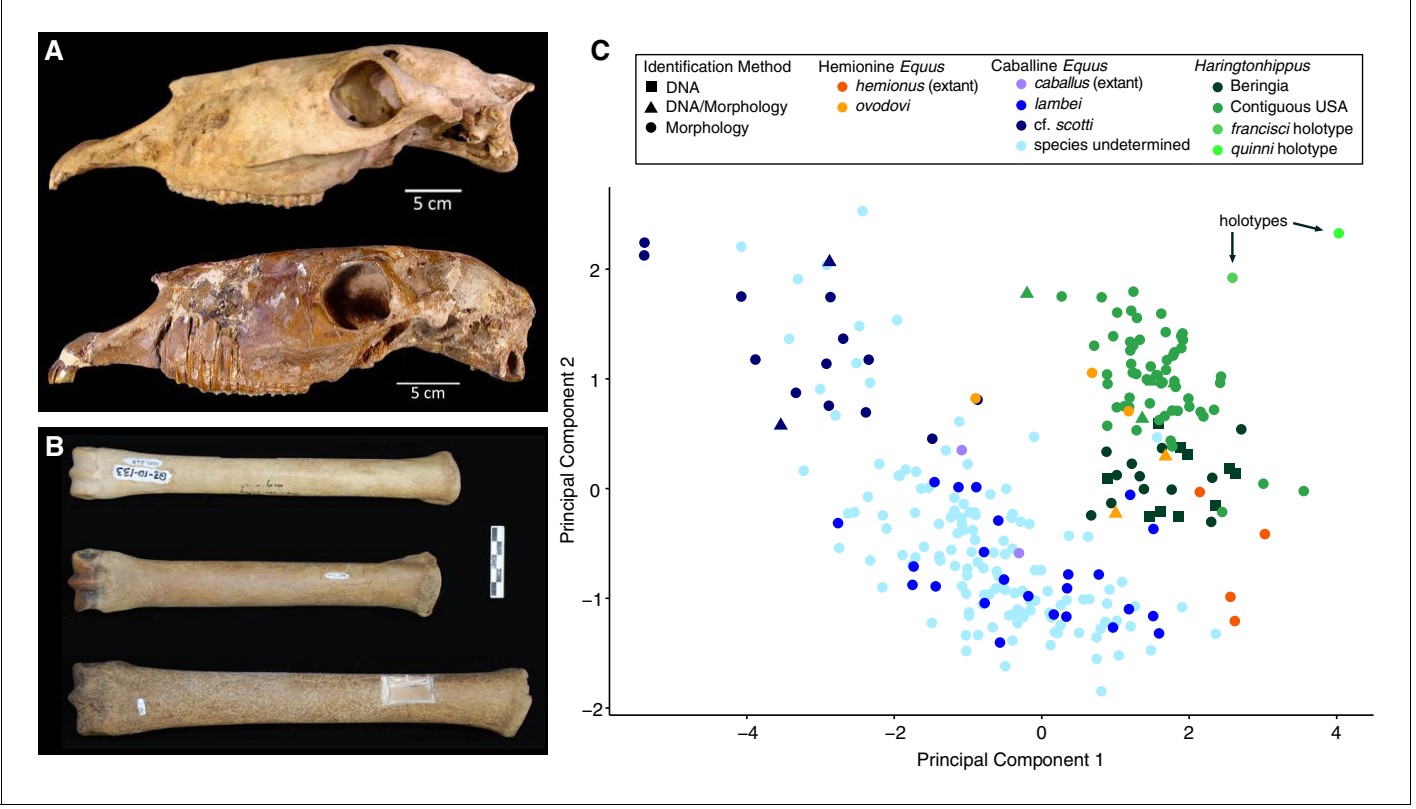

**Figure 2.** Morphological analysis of extant and middle-late Pleistocene equids. (**A**) Crania of *Haringtonhippus francisci*, upper: LACM(CIT) 109/156450 from Nevada, lower: TMM 34–2518 from Texas. (**B**) From upper to lower, third metatarsals of: *H. francisci* (YG 401.268), *E. lambei* (YG 421.84), and *E.* cf. *scotti* (YG 198.1) from Yukon. Scale bar is 5 cm. (**C**) Principal component analysis of selected third metatarsals from extant and middle-late Pleistocene equids, showing clear clustering of stilt-legged (hemionine *Equus* (orange) and *H. francisci* (green)) from stout-legged (caballine *Equus*; blue) specimens (see also *Figure 2—source data 1*). Symbol shape denotes the specimen identification method (DNA: square, triangle: DNA/morphology, circle: morphology). The first and second principal components explain 95% of the variance.

DOI: https://doi.org/10.7554/eLife.29944.010

The following source data and figure supplements are available for figure 2:

**Source data 1.** Measurement data for (**A**) equid third metatarsals, which were used in the morphometrics analysis, and (**B**) other NWSL equid elements.
DOI: https://doi.org/10.7554/eLife.29944.015

**Figure supplement 1.** The two crania assigned to *H. francisci*.
DOI: https://doi.org/10.7554/eLife.29944.011

**Figure supplement 2.** Comparison between the limb bones of *H. francisci*, *E. lambei*, and *E.* cf. *scotti* from Yukon.
DOI: https://doi.org/10.7554/eLife.29944.012

**Figure supplement 3.** An example equid metacarpal from Natural Trap Cave, Wyoming.
DOI: https://doi.org/10.7554/eLife.29944.013

**Figure supplement 4.** An example femur of *H. francisci* from Gypsum Cave, Nevada.
DOI: https://doi.org/10.7554/eLife.29944.014

## Etymology

The new genus is named in honor of C. Richard Harington, who first described NWSL equids from eastern Beringia (*Harington and Clulow, 1973*). '*Hippus*' is from the Greek word for horse, and so *Haringtonhippus* is implied to mean 'Harington's horse'.

## Holotype

A partial skeleton consisting of a complete cranium, mandible, and a stilt-legged third metatarsal (MTIII) (*Figure 2a* and *Figure 2—figure supplement 1b*), which is curated at the Texas Vertebrate Paleontology Collections at The University of Texas, Austin (TMM 34–2518). This specimen is the

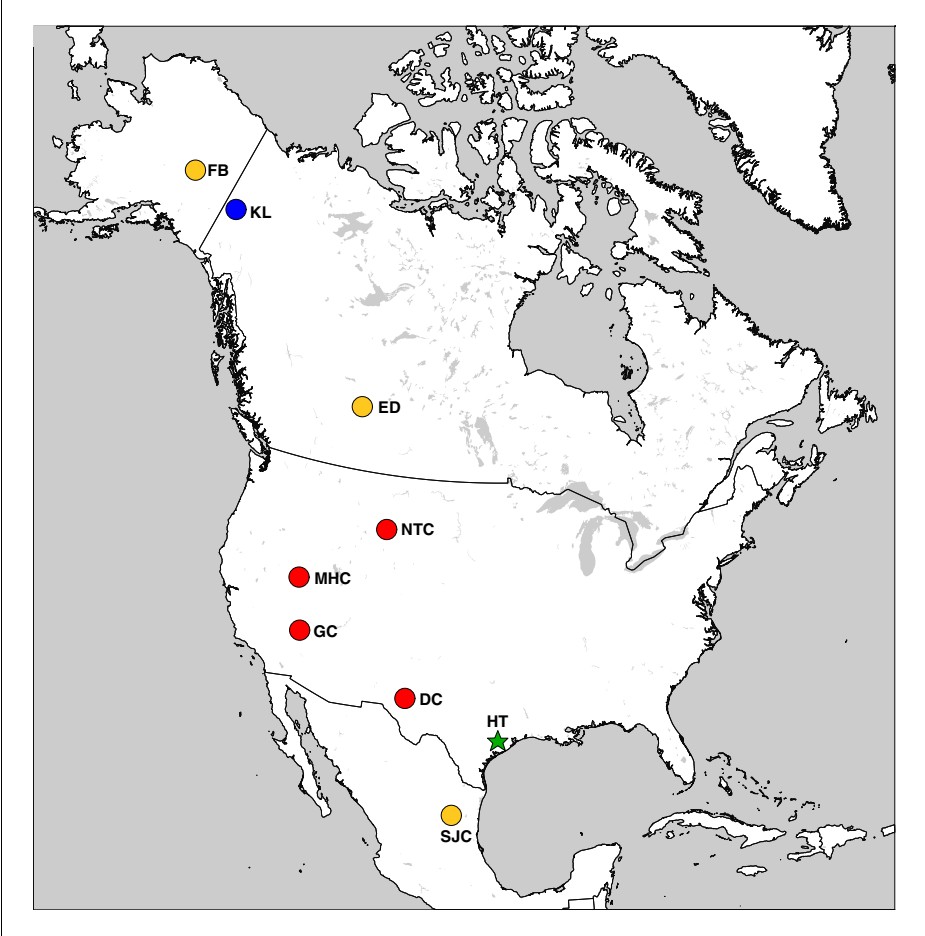

**Figure 3.** The geographic distribution of *Haringtonhippus*. Blue circles are east Beringian localities (KL: Klondike region, Yukon Territory, Canada). Red circles are contiguous USA localities (NTC: Natural Trap Cave, Wyoming, USA; GC: Gypsum Cave, Nevada, USA; MHC: Mineral Hill Cave, Nevada, USA; DC: Dry Cave, New Mexico, USA [*Barrón-Ortiz et al., 2017*; *Weinstock et al., 2005*]). Orange circles are localities with tentatively assigned *Haringtonhippus* specimens only (FB: Fairbanks, Alaska, USA; ED: Edmonton, Alberta, Canada, USA; SJC: San Josecito Cave, Nuevo Leon, Mexico; (*Barrón-Ortiz et al., 2017*; *Guthrie, 2003*). The green-star-labeled HT is the locality of the *francisci* holotype, Wharton County, Texas, USA. This figure was drawn using Simplemappr (*Shorthouse, 2010*).

DOI: https://doi.org/10.7554/eLife.29944.016

holotype of '*E*'. *francisci*, originally described by *Hay (1915)*, and is from the middle Pleistocene Lissie Formation of Wharton County, Texas (*Hay, 1915*; *Lundelius and Stevens, 1970*).

## Referred material

On the basis of mitochondrial and nuclear genomic data, we assign the following material confidently to *Haringtonhippus*: a cranium, femur, and MTIII (LACM(CIT): Nevada); three MTIIIs, three third metacarpals (MCIII), three premolar teeth, and a molar tooth (KU: Wyoming); two radii, 12 MTIIIs, three MCIIIs, a metapodial, and a first phalanx (YG: Yukon Territory); and a premolar tooth (University of Texas El Paso, UTEP: New Mexico); (*Figure 2—figure supplements 1–4* and *Supplementary file 1*; (*Barrón-Ortiz et al., 2017*; *Weinstock et al., 2005*). This material includes at least four males and at least six females (Appendix 2, *Appendix 2—Table 4* and Appendix 2—Table 4—source data 1). We further assign MTIII specimens from Yukon Territory (n = 13), Wyoming (n = 57), and Nevada (n = 4) to *Haringtonhippus* on the basis of morphometric analysis (*Figure 2c* and *Figure 2—source data 1*). On the basis of short mitochondrial DNA sequences, we tentatively assign to *Haringtonhippus* a premolar tooth (LACM(CIT): Nuevo Leon); a premolar and a molar (UTEP: New Mexico); and a premolar (Royal Alberta Museum, RAM/PMA: Alberta) (*Barrón-*

*Ortiz et al., 2017*). We also tentatively assign 19 NWSL equid metapodial specimens from the Fairbanks area, Alaska (*Guthrie, 2003*) to *Haringtonhippus*, but note that morphometric and/or palaeogenomic analysis would be required to confirm this designation.

## Geographic and temporal distribution

*Haringtonhippus* is known only from the Pleistocene of North America (*Figure 3*). In addition to the middle Pleistocene holotype from Texas, *Haringtonhippus* is confidently known from the late Pleistocene of Yukon Territory (Klondike region), Wyoming (Natural Trap Cave), Nevada (Gypsum Cave, Mineral Hill Cave), and New Mexico (Dry Cave), and is tentatively registered as present in Nuevo Leon (San Josecito Cave), Alberta (Edmonton area), and Alaska (Fairbanks area) (Appendix 2, *Supplementary file 1*, and *Appendix 2—table 3*; [*Barrón-Ortiz et al., 2017*; *Vilstrup et al., 2013*; *Weinstock et al., 2005*]).

To investigate the last appearance date (LAD) of *Haringtonhippus* in eastern Beringia, we obtained new radiocarbon dates from 17 Yukon Territory fossils (Appendix 1 and *Supplementary file 1*). This resulted in three statistically-indistinguishable radiocarbon dates of ~14.4 $^{14}$C ka BP (derived from two independent laboratories) from a metacarpal bone (YG 401.235) of *Haringtonhippus*, which represents this taxon's LAD in eastern Beringia (*Supplementary file 1*). The LAD for North America as a whole is based on two dates of ~13.1 $^{14}$C ka BP from Gypsum Cave, Nevada (*Supplementary file 1*; [*Weinstock et al., 2005*]).

## Mitogenomic diagnosis

*Haringtonhippus* is the sister genus to *Equus* (equid crown group), with *Hippidion* being sister to the *Haringtonhippus-Equus* clade (*Figure 1*). *Haringtonhippus* can be differentiated from *Equus* and *Hippidion* by 178 synapomorphic positions in the mitochondrial genome, including four insertions and 174 substitutions (*Appendix 1—Table 2* and *Appendix 1—table 2—source data 1*). We caution that these synapomorphies are tentative and will likely be reduced in number as a greater diversity of mitochondrial genomes for extinct equids become available.

## Morphological comparisons of third metatarsals

We used morphometric analysis of caballine/stout-legged *Equus* and stilt-legged equids (hemionine/stilt-legged *Equus*, *Haringtonhippus*) MTIIIs to determine how confidently these groups can be distinguished (*Figure 2c*). Using logistic regression on principal components, we find a strong separation that can be correctly distinguished with 98.2% accuracy (Appendix 2; *Heintzman et al., 2017*). Hemionine/stilt-legged *Equus* MTIIIs occupy the same morphospace as *H. francisci* in our analysis, although given a larger sample size, it may be possible to discriminate *E. hemionus* from the remaining stilt-legged equids. We note that *Haringtonhippus* seems to exhibit a negative correlation between latitude and MTIII length, and that specimens from the same latitude occupy similar morphospace regardless of whether DNA- or morphological-based identification was used (*Figure 2c* and *Figure 2—source data 1*).

## Comments

On the basis of morphology, we assign all confidently referred material of *Haringtonhippus* to the single species *H. francisci Hay (1915)* (Appendix 2). Comparison between the cranial anatomical features of LACM(CIT) 109/156450 and TMM 34–2518 reveal some minor differences, which can likely be ascribed to intraspecific variation (*Figure 2a* and Appendix 2 and *Figure 2—figure supplement 1*). Further, the MTIII of TMM 34–2518 is comparable to the MTIIIs ascribed to *Haringtonhippus* by palaeogenomic data, and is consistent with the observed latitudinally correlated variation in MTIII length across *Haringtonhippus* (*Figure 2c* and Appendix 2).

This action is supported indirectly by molecular evidence, namely the lack of mitochondrial phylogeographic structure and the estimated time to most recent common ancestor (tMRCA) for sampled *Haringtonhippus*. The mitochondrial tree topology within *Haringtonhippus* does not exhibit phylogeographic structure (*Figure 1—figure supplement 1b*), which is consistent with sampled *Haringtonhippus* mitochondrial genomes belonging to the same species. Using Bayesian time-tree analysis, we estimated a tMRCA for the sampled *Haringtonhippus* mitochondrial genomes of ~200–470 ka BP (*Figure 1* and *Figure 1—source data 1*; *Heintzman et al., 2017*). The MRCA of *Haringtonhippus* is

therefore more recent than that of other extant equid species (such as *E. asinus* and *E. quagga*, which have a combined 95% HPD range: 410–1030 ka BP; *Figure 1* and *Figure 1—source data 1*; *Heintzman et al., 2017*). Although the middle Pleistocene holotype TMM 34–2518 (~125–780 ka BP) may predate our *Haringtonhippus* mitochondrial tMRCA, this sample has no direct date and the range of possible ages falls within the tMRCA range of other extant equid species. We therefore cannot reject the hypothesis of its conspecificity with *Haringtonhippus,* as defined palaeogenomically. We attempted, but were unable, to recover either collagen or genomic data from TMM 34–2518 (Appendix 2), consistent with the taphonomic, stratigraphic, and geographic context of this fossil (*Hay, 1915*; *Lundelius and Stevens, 1970*). Altogether, the molecular evidence is consistent with the assignment of *H. francisci* as the type and only species of *Haringtonhippus*.

## Discussion

### Reconciling the genomic and fossil records of Plio-Pleistocene equid evolution

The suggested placement of NWSL equids within a taxon (*Haringtonhippus*) sister to *Equus* is a departure from previous interpretations, which variably place the former within *Equus*, as sister to hemionines or caballine horses (*Figure 1*). According to broadly accepted palaeontological interpretations, the earliest equids exhibiting morphologies consistent with NWSL and caballine attribution appear in the fossil record only ~2–3 and ~1.9–0.7 Ma ago (*Eisenmann et al., 2008*; *Forsten, 1992*), respectively, whereas our divergence estimates suggest that these lineages to have diverged between 4.1–5.8 and 3.8–4.5 Ma, most likely in North America. Dating incongruence might be attributed to an incomplete fossil record, but this seems unlikely given the density of the record for late Neogene and Pleistocene horses. Conversely, incongruence might be attributed to problems with estimating divergence using genomic evidence. However, we emphasize that the NWSL-*Equus* split is robustly calibrated to the caballine/non-caballine *Equus* divergence at 4.0–4.5 Ma, which is in turn derived from a direct molecular clock calibration using a middle Pleistocene horse genome (*Orlando et al., 2013*).

Other possibilities to explain the incongruence include discordance between the timing of species divergence and the evolution of diagnostic anatomical characteristics, or failure to detect or account for homoplasy (*Forsten, 1992*). For example, Pliocene *Equus* generally exhibits a primitive ('plesippine' in North America, 'stenonid' in the Old World) morphology that presages living zebras and asses (*Forsten, 1988*, *1992*), with more derived caballine (stout-legged) and hemionine (stilt-legged) forms evolving in the early Pleistocene. The stilt-legged morphology appears to have evolved independently at least once in each of the Old and New Worlds, yielding the Asiatic wild asses and *Haringtonhippus*, respectively. We include the middle-late Pleistocene Eurasian *E. hydruntinus* within the Asiatic wild asses (following [*Bennett et al., 2017*; *Burke et al., 2003*; *Orlando et al., 200*6]), and note that the Old World sussemione *E. ovodovi* may represent another instance of independent stilt-legged origin, but its relation to Asiatic wild asses and other non-caballine *Equus* is currently unresolved (as depicted in *Der Sarkissian et al., 2015*; *Orlando et al., 2009*; *Vilstrup et al., 2013*; and *Figure 1*). It is plausible that features at the plesiomorphous end of the spectrum, such as those associated with *Hippidion*, survived after the early to middle Pleistocene at lower latitudes (South America, Africa; *Figure 1*). By contrast, the more derived hemionine and caballine morphologies evolved from, and replaced, their antecedents in higher latitude North America and Eurasia, perhaps as adaptations to the extreme ecological pressures perpetuated by the advance and retreat of continental ice sheets and correlated climate oscillations during the Pleistocene (*Forsten, 1992*, Forsten, 1996*Forsten, 1996*). We note that this high-latitude replacement model is consistent with the turnover observed in regional fossil records for Pleistocene equids in North America (*Azzaroli, 1992*; *Azzaroli and Voorhies, 1993*) and Eurasia (*Forsten, 1988*, *1992*, *Forsten, 1996*). By contrast, in South America *Hippidion* co-existed with caballine horses until they both succumbed to extinction, together with much of the New World megafauna near the end of the Pleistocene (*Forsten, 1996*; *Koch and Barnosky, 2006*; *O'Dea et al., 2016*). This model helps to explain the discordance between the timings of the appearance of the caballine and hemionine morphologies in the fossil record and the divergence of lineages leading to these forms as estimated from palaeogenomic data.

Although we can offer no solution to the general problem of mismatches between molecular and morphological divergence estimators–an issue scarcely unique to equid systematics–this model predicts that some previously described North American Pliocene and early Pleistocene *Equus* species (e.g. *E. simplicidens*, *E. idahoensis*; [*Azzaroli and Voorhies, 1993*]), or specimens thereof, may be ancestral to extant *Equus* and/or late Pleistocene *Haringtonhippus*.

## Temporal and geographic range overlap of Pleistocene equids in North America

Three new radiocarbon dates of ~14.4 $^{14}$C ka BP from a Yukon *Haringtonhippus* fossil greatly extends the known temporal range of this genus in eastern Beringia. This result demonstrates, contrary to its previous LAD of 31,400 ± 1200 $^{14}$C years ago (AA 26780; [*Guthrie, 2003*]), that *Haringtonhippus* survived throughout the last glacial maximum in eastern Beringia (*Clark et al., 2009*) and may have come into contact with humans near the end of the Pleistocene (*Goebel et al., 2008*; *Guthrie, 2006*). These data suggest that populations of stilt-legged *Haringtonhippus* and stout-legged caballine *Equus* were sympatric, both north and south of the continental ice sheets, through the late Pleistocene and became extinct at roughly the same time. The near synchronous extinction of both horse groups across their entire range in North America suggests that similar causal mechanisms may have led each to their demise.

The sympatric nature of these equids raises questions of whether they managed to live within the same community without hybridizing or competing for resources. Extant members of the genus *Equus* vary considerably in the sequence of Prdm9, a gene involved in the speciation process, and chromosome number (karyotype) (*Ryder et al., 1978*; *Steiner and Ryder, 2013*), and extant caballine and non-caballine *Equus* rarely produce fertile offspring (*Allen and Short, 1997*; *Steiner and Ryder, 2013*). It is unlikely, therefore, that the more deeply diverged *Haringtonhippus* and caballine *Equus* would have been able to hybridize. Future analysis of high coverage nuclear genomes, ideally including an outgroup such as *Hippidion*, will make it possible to test for admixture that may have occurred soon after the lineages leading to *Haringtonhippus* and *Equus* diverged, as occurred between the early caballine and non-caballine *Equus* lineages (*Jónsson et al., 2014*). It may also be possible to use isotopic and/or tooth mesowear analyses to assess the potential of resource partitioning between *Haringtonhippus* and caballine *Equus* in the New World.

## Fossil systematics in the palaeogenomics and proteomics era: concluding remarks

Fossils of NWSL equids have been known for more than a century, but until the present study their systematic position within Plio-Pleistocene Equidae was poorly characterized. This was not because of a lack of interest on the part of earlier workers, whose detailed anatomical studies strongly indicated that what we now call *Haringtonhippus* was related to Asiatic wild asses, such as Tibetan khulan and Persian onagers, rather than to caballine horses (*Eisenmann et al., 2008*; *Guthrie, 2003*; *Scott, 2004*; *Skinner and Hibbard, 1972*). That the cues of morphology have turned out to be misleading in this case underlines a recurrent problem in systematic biology, which is how best to discriminate authentic relationships within groups, such as Neogene equids, that were prone to rampant convergence. The solution we adopted here was to utilize both palaeogenomic and morphometric information in reframing the position of *Haringtonhippus*, which now clearly emerges as the closest known outgroup to all living *Equus*.

Our success in this regard demonstrates that an approach which incorporates phenomics with molecular methods (palaeogenomic as well as palaeoproteomic, e.g. [*Welker et al., 2015*]) is likely to offer a means for securely detecting relationships within speciose groups that are highly diverse ecomorphologically. All methods have their limits, with taphonomic degradation being the critical one for molecular approaches. However, proteins may persist significantly longer than ancient DNA (e.g. [*Rybczynski et al., 2013*]), and collagen proteomics may come to play a key role in characterizing affinities, as well as the reality, of several proposed Neogene equine taxa (e.g. *Dinohippus*, *Pliohippus*, *Protohippus*, *Calippus*, and *Astrohippus*; [*MacFadden, 1998*]) whose distinctiveness and relationships are far from settled (*Azzaroli and Voorhies, 1993*; *Forsten, 1992*). A reciprocally informative approach like the one taken here holds much promise for lessening the amount of systematic noise, due to oversplitting, that hampers our understanding of the evolutionary biology of other

major late Pleistocene megafaunal groups such as bison and mammoths (*Enk et al., 2016*; *Froese et al., 2017*). This approach is clearly capable of providing new insights into just how extensive megafaunal losses were at the end of the Pleistocene, in what might be justifiably called the opening act of the Sixth Mass Extinction in North America.

## Materials and methods

We provide an overview of methods here; full details can be found in Appendix 1.

### Sample collection and radiocarbon dating

We recovered Yukon fossil material (17 *Haringtonhippus francisci*, two *Equus* cf. *scotti*, and two *E. lambei*; *Supplementary file 1*) from active placer mines in the Klondike goldfields near Dawson City. We further sampled seven *H. francisci* fossils from the contiguous USA that are housed in collections at the University of Kansas Biodiversity Institute (KU; n = 4), Los Angeles County Museum of Natural History (LACM(CIT); n = 2), and the Texas Vertebrate Paleontology Collections at The University of Texas (TMM; n = 1). We radiocarbon dated the Klondike fossils and the *H. francisci* cranium from the LACM(CIT) (*Supplementary file 1*).

### Morphometric analysis of third metatarsals

For morphometric analysis, we took measurements of third metatarsals (MTIII) and other elements. We used a reduced data set of four MTIII variables for principal components analysis and performed logistic regression on the first three principal components, computed in R (*R Development Core Team, 2008*) (Source code 1).

### DNA extraction, library preparation, target enrichment, and sequencing

We conducted all molecular biology methods prior to indexing PCR in the dedicated palaeogenomics laboratory facilities at either the UC Santa Cruz or Pennsylvania State University. We extracted DNA from between 100 and 250 mg of bone powder following either *Rohland et al. (2010)* or *Dabney et al. (2013a)*. We then converted DNA extracts to libraries following the Meyer and Kircher protocol (*Meyer and Kircher, 2010*), as modified by (*Heintzman et al., 2015*) or the PSU method of (*Vilstrup et al., 2013*). We enriched libraries for equid mitochondrial DNA. We then sequenced all enriched libraries and unenriched libraries from 17 samples using Illumina platforms.

### Mitochondrial genome reconstruction and analysis

We prepared raw sequence data for alignment and mapped the filtered reads to the horse reference mitochondrial genome (Genbank: NC_001640.1) and a *H. francisci* reference mtDNA genome (Genbank: KT168321), resulting in mitogenomic coverage ranging from $5.8\times$ to $110.7\times$ (*Supplementary file 1*). We were unable to recover equid mtDNA from TMM 34–2518 (the *francisci* holotype) using this approach (Appendix 2). We supplemented our mtDNA genome sequences with 38 previously published complete equid mtDNA genomes. We constructed six alignment data sets and selected models of molecular evolution for the analyses described below (*Appendix 1—table 1*, and *Supplementary file 1*; *Heintzman et al., 2017*).

We tested the phylogenetic position of the NWSL equids (=*H. francisci*) using mtDNA data sets 1–3 and applying Bayesian (*Ronquist et al., 2012*) and maximum likelihood (ML; [*Stamatakis, 2014*]) analyses. We varied the outgroup, the inclusion or exclusion of the fast-evolving partitions, and the inclusion or exclusion of *Hippidion* sequences. Due to the lack of a globally supported topology across the Bayesian and ML phylogenetic analyses, we used an Evolutionary Placement Algorithm (EPA; [*Berger et al., 2011*]) to determine the *a posteriori* likelihood of phylogenetic placements for candidate equid outgroups using mtDNA data set four. We also used the same approach to assess the placement of previously published equid sequences (Appendix 2). To infer divergence times between the four major equid groups (*Hippidion*, NWSL equids, caballine *Equus*, and non-caballine *Equus*), we ran Bayesian timetree analyses (*Drummond et al., 2012*) using mtDNA data set five. We varied these analyses by including or excluding fast-evolving partitions, constrained the root height or not, and including or excluding the *E. ovodovi* sequence.

To facilitate future identification of equid mtDNA sequences, we constructed, using data set six, a list of putative synapomorphic base states, including indels and substitutions, that define the genera *Hippidion*, *Haringtonhippus*, and *Equus* at sites across the mtDNA genome.

## Phylogenetic inference, divergence date estimation, and sex determination from nuclear genomes

To test whether our mtDNA genome-based phylogenetic hypothesis truly reflects the species tree, we compared the nuclear genomes of a horse (EquCab2), donkey (*Orlando et al., 2013*), and the shotgun sequence data from 17 of our NWSL equid samples (*Figure 1—source data 2*, Appendix 1, *Appendix 1—figure 1*, and *Supplementary file 1*). We applied four successive approaches, which controlled for reference genome and DNA fragment length biases (Appendix 1).

We estimated the divergence between the NWSL equids and *Equus* (horse and donkey) by fitting the branch length, or relative private transversion frequency, ratio between horse/donkey and NWSL equids into a simple phylogenetic scenario (*Figure 1—figure supplement 3*). We then multiplied the NWSL equid branch length by a previous horse-donkey divergence estimate (4.0–4.5 Ma; [*Orlando et al., 2013*]) to give the estimated NWSL equid-*Equus* divergence date, following (*Heintzman et al., 2015*) and assuming a strict genome-wide molecular clock (*Figure 1—figure supplement 3*).

We determined the sex of the 17 NWSL equid samples by comparing the relative mapping frequency of the autosomes to the X chromosome.

## DNA damage analysis

We assessed the prevalence of mitochondrial and nuclear DNA damage in a subset of the equid samples using mapDamage (*Jónsson et al., 2013*).

## Data availability

Repository details and associated metadata for curated samples can be found in *Supplementary file 1*. MTIII and other element measurement data are in *Figure 2—source data 1*, and the Rscript used for morphometric analysis is in the DRYAD database (*Heintzman et al., 2017*). MtDNA genome sequences have been deposited in Genbank under accessions KT168317-KT168336, MF134655-MF134663, and an updated version of JX312727. All mtDNA genome alignments (in NEXUS format) and associated XML and TREE files are in the DRYAD database (*Heintzman et al., 2017*). Raw shotgun sequence data used for the nuclear genomic analyses and raw shotgun and target enrichment sequence data for TMM 34–2518 (*francisci* holotype) have been deposited in the Short Read Archive (BioProject: PRJNA384940).

## Nomenclatural act

The electronic edition of this article conforms to the requirements of the amended International Code of Zoological Nomenclature, and hence the new name contained herein is available under that Code from the electronic edition of this article. This published work and the nomenclatural act it contains have been registered in ZooBank, the online registration system for the ICZN. The ZooBank LSIDs (Life Science Identifiers) can be resolved and the associated information viewed through any standard web browser by appending the LSID to the prefix 'http://zoobank.org/'. The LSID for this publication is: urn:lsid:zoobank.org:pub:8D270E0A-9148-4089-920C-724F07D8DC0B. The electronic edition of this work was published in a journal with an ISSN, and has been archived and is available from the following digital repositories: PubMed Central and LOCKSS.

## Acknowledgements

We thank the Klondike placer gold mining community of Yukon for their support and providing access to their mines from which many of our *Haringtonhippus* fossils were recovered. We thank Matt Brown and Chris Sagebiel of the Texas Vertebrate Palaeontology Collections at the University of Texas, Austin for access to a portion of TMM 34–2518, and also thank Sam McLeod, Vanessa Rhue, and Aimee Montenegro at the Los Angeles County Museum for access to the Gypsum Cave material for consumptive sampling. Thanks to Brent Breithaupt (Bureau of Land Management) for

permitting the sampling of fossils from Natural Trap Cave that were originally recovered by Larry Martin, Miles Gilbert, and colleagues, and are presently curated by the University of Kansas Biodiversity Institute. We thank Chris Beard and David Burnham (University of Kansas) for facilitating access to these fossils. Thanks to Tom Guilderson, Andrew Fields, Dan Chang, and Samuel Vohr for technical assistance. Thanks to Greger Larson for providing the base map in *Figure 1*. We thank the reviewers whose comments improved this manuscript. This work used the Vincent J Coates Genomics Sequencing Laboratory at UC Berkeley, supported by NIH S10 Instrumentation Grants S10RR029668 and S10RR027303. PDH, JAC, JDK, MS, and BS were supported by NSF grants PLR-1417036 and 09090456, and Gordon and Betty Moore Foundation Grant GBMF3804. PDH received support from Norway's Research Council (Grant 250963: 'ECOGEN'). LO was supported by the Danish Council for Independent Research Natural Sciences (Grant 4002-00152B); the Danish National Research Foundation (Grant); the 'Chaires d'Attractivit. 2014' IDEX, University of Toulouse, France (OURASI), and the European Research Council (ERC-CoG-2015–681605).

## Additional information

### Funding

| Funder | Grant reference number | Author |
| --- | --- | --- |
| National Science Foundation | PLR-1417036 | Peter D Heintzman<br>James A Cahill<br>Joshua D Kapp<br>Mathias Stiller<br>Beth Shapiro |
| National Science Foundation | PLR-09090456 | Peter D Heintzman<br>James A Cahill<br>Joshua D Kapp<br>Mathias Stiller<br>Beth Shapiro |
| Danish Council for Independent Research Natural Sciences | 4002-00152B | Ludovic Orlando |
| Danmarks Grundforskningsfond | | Ludovic Orlando |
| European Research Council | ERC-CoG-2015-681605 | Ludovic Orlando |
| Gordon and Betty Moore Foundation | GBMF3804 | Peter D Heintzman<br>James A Cahill<br>Joshua D Kapp<br>Mathias Stiller<br>Beth Shapiro |
| Norges Forskningsråd | 250963 | Peter D Heintzman |

The funders had no role in study design, data collection and interpretation, or the decision to submit the work for publication.

### Author contributions

Peter D Heintzman, Conceptualization, Data curation, Software, Formal analysis, Supervision, Validation, Investigation, Visualization, Methodology, Writing—original draft, Project administration, Writing—review and editing; Grant D Zazula, Conceptualization, Resources, Investigation, Writing—original draft, Writing—review and editing; Ross DE MacPhee, Eric Scott, Resources, Validation, Investigation, Writing—original draft, Writing—review and editing; James A Cahill, Software, Formal analysis, Visualization, Writing—original draft; Brianna K McHorse, Data curation, Software, Formal analysis, Visualization, Methodology, Writing—original draft; Joshua D Kapp, Validation, Investigation; Mathias Stiller, Resources, Investigation, Methodology; Matthew J Wooller, Duane G Froese, Resources, Funding acquisition, Writing—review and editing; Ludovic Orlando, Resources, Writing—review and editing; John Southon, Resources, Investigation; Beth Shapiro, Conceptualization, Resources, Supervision, Funding acquisition, Writing—original draft, Project administration, Writing—review and editing

## Author ORCIDs

Peter D Heintzman (iD) http://orcid.org/0000-0002-6449-0219
Ross DE MacPhee (iD) https://orcid.org/0000-0003-0688-0232
Eric Scott (iD) http://orcid.org/0000-0002-2730-0893
James A Cahill (iD) https://orcid.org/0000-0002-7145-0215
Beth Shapiro (iD) https://orcid.org/0000-0002-2733-7776

## Ethics

We received permission from three entities to destructively sample palaeontological specimens: the Texas Vertebrate Paleontology Collections at The University of Texas (granted to PDH and ES), the Los Angeles County Museum (granted to ES), and the US Department of the Interior Bureau of Land Management, Wyoming (granted to RDEM and BS; reference number: 8270(930))

## Decision letter and Author response

Decision letter https://doi.org/10.7554/eLife.29944.096
Author response https://doi.org/10.7554/eLife.29944.097

# Additional files

## Supplementary files

• Supplementary file 1. Metadata for all samples used in the mitochondrial and nuclear genomic analyses, with the *francisci* holotype included for reference. *mtDNA coverage is based on the iterative assembler or as previously published. **New mtDNA genome sequence, coverage, and radiocarbon data are reported for MS272.
DOI: https://doi.org/10.7554/eLife.29944.017

• Transparent reporting form
DOI: https://doi.org/10.7554/eLife.29944.018

## Major datasets

The following datasets were generated:

| Author(s) | Year | Dataset title | Dataset URL | Database, license, and accessibility information |
|---|---|---|---|---|
| Heintzman PD, Cahill JA, Kapp JD, Stiller M, Shapiro B | 2017 | Nuclear DNA sequences from 17 *Haringtonhippus francisci* fossils | https://www.ncbi.nlm.nih.gov/bioproject/PRJNA384940 | Publicly available at NCBI Short Read Archive (accession no: PRJNA384940) |
| Heintzman PD, Shapiro B | 2017 | Mitochondrial genome sequence from YG 303.371 | https://www.ncbi.nlm.nih.gov/nuccore/KT168317 | Publicly available at NCBI GenBank (accession no: KT168317) |
| Heintzman PD, Stiller M, Shapiro B | 2017 | Mitochondrial genome sequence from YG 133.16 | https://www.ncbi.nlm.nih.gov/nuccore/KT168318 | Publicly available at NCBI GenBank (accession no: KT168318) |
| Heintzman PD, Shapiro B | 2017 | Mitochondrial genome sequence from YG 29.169 | https://www.ncbi.nlm.nih.gov/nuccore/KT168319 | Publicly available at NCBI GenBank (accession no: KT168319) |
| Heintzman PD, Stiller M, Shapiro B | 2017 | Mitochondrial genome sequence from YG 401.387 | https://www.ncbi.nlm.nih.gov/nuccore/KT168320 | Publicly available at NCBI GenBank (accession no: KT168320) |
| Heintzman PD, Shapiro B | 2017 | Mitochondrial genome sequence from YG 404.663 | https://www.ncbi.nlm.nih.gov/nuccore/KT168321 | Publicly available at NCBI GenBank (accession no: KT168321) |

| | | | | |
|---|---|---|---|---|
| Heintzman PD, Stiller M, Shapiro B | 2017 | Mitochondrial genome sequence from YG 328.54 | https://www.ncbi.nlm.nih.gov/nuccore/KT168322 | Publicly available at NCBI GenBank (accession no: KT168322) |
| Heintzman PD, Shapiro B | 2017 | Mitochondrial genome sequence from YG 378.5 | https://www.ncbi.nlm.nih.gov/nuccore/KT168323 | Publicly available at NCBI GenBank (accession no: KT168323) |
| Heintzman PD, Shapiro B | 2017 | Mitochondrial genome sequence from YG 404.478 | https://www.ncbi.nlm.nih.gov/nuccore/KT168324 | Publicly available at NCBI GenBank (accession no: KT168324) |
| Heintzman PD, Shapiro B | 2017 | Mitochondrial genome sequence from YG 402.235 | https://www.ncbi.nlm.nih.gov/nuccore/KT168325 | Publicly available at NCBI GenBank (accession no: KT168325) |
| Heintzman PD, Stiller M, Shapiro B | 2017 | Mitochondrial genome sequence from YG 130.55 | https://www.ncbi.nlm.nih.gov/nuccore/KT168326 | Publicly available at NCBI GenBank (accession no: KT168326) |
| Heintzman PD, Shapiro B | 2017 | Mitochondrial genome sequence from YG 198.1 | https://www.ncbi.nlm.nih.gov/nuccore/KT168327 | Publicly available at NCBI GenBank (accession no: KT168327) |
| Heintzman PD, Stiller M, Shapiro B | 2017 | Mitochondrial genome sequence from YG 303.1085 | https://www.ncbi.nlm.nih.gov/nuccore/KT168328 | Publicly available at NCBI GenBank (accession no: KT168328) |
| Heintzman PD, Shapiro B | 2017 | Mitochondrial genome sequence from YG 130.6 | https://www.ncbi.nlm.nih.gov/nuccore/KT168329 | Publicly available at NCBI GenBank (accession no: KT168329) |
| Heintzman PD, Shapiro B | 2017 | Mitochondrial genome sequence from YG 417.13 | https://www.ncbi.nlm.nih.gov/nuccore/KT168330 | Publicly available at NCBI GenBank (accession no: KT168330) |
| Heintzman PD, Shapiro B | 2017 | Mitochondrial genome sequence from YG 76.2 | https://www.ncbi.nlm.nih.gov/nuccore/KT168331 | Publicly available at NCBI GenBank (accession no: KT168331) |
| Heintzman PD, Shapiro B | 2017 | Mitochondrial genome sequence from YG 160.8 | https://www.ncbi.nlm.nih.gov/nuccore/KT168332 | Publicly available at NCBI GenBank (accession no: KT168332) |
| Heintzman PD, Shapiro B | 2017 | Mitochondrial genome sequence from YG 404.662 | https://www.ncbi.nlm.nih.gov/nuccore/KT168333 | Publicly available at NCBI GenBank (accession no: KT168333) |
| Heintzman PD, Shapiro B | 2017 | Mitochondrial genome sequence from YG 404.480 | https://www.ncbi.nlm.nih.gov/nuccore/KT168334 | Publicly available at NCBI GenBank (accession no: KT168334) |
| Heintzman PD, Stiller M, Shapiro B | 2017 | Mitochondrial genome sequence from YG 401.235 | https://www.ncbi.nlm.nih.gov/nuccore/KT168335 | Publicly available at NCBI GenBank (accession no: KT168335) |
| Heintzman PD, Shapiro B | 2017 | Mitochondrial genome sequence from YG 404.205 | https://www.ncbi.nlm.nih.gov/nuccore/KT168336 | Publicly available at NCBI GenBank (accession no: KT168336) |
| Heintzman PD, Kapp JD, Shapiro B | 2017 | Mitochondrial genome sequence from LACM(CIT) 109 / 150807 | https://www.ncbi.nlm.nih.gov/nuccore/MF134655 | Publicly available at NCBI GenBank (accession no: MF134655) |

| Heintzman PD, Kapp JD, Shapiro B | 2017 | Mitochondrial genome sequence from LACM(CIT) 109 / 149291 | https://www.ncbi.nlm.nih.gov/nuccore/MF134656 | Publicly available at NCBI GenBank (accession no: MF134656) |
|---|---|---|---|---|
| Heintzman PD, Kapp JD, Shapiro B | 2017 | Mitochondrial genome sequence from LACM(CIT) 109 / 156450 | https://www.ncbi.nlm.nih.gov/nuccore/MF134657 | Publicly available at NCBI GenBank (accession no: MF134657) |
| Heintzman PD, Kapp JD, Shapiro B | 2017 | Mitochondrial genome sequence from KU 47800 | https://www.ncbi.nlm.nih.gov/nuccore/MF134658 | Publicly available at NCBI GenBank (accession no: MF134658) |
| Heintzman PD, Kapp JD, Shapiro B | 2017 | Mitochondrial genome sequence from KU 62055 | https://www.ncbi.nlm.nih.gov/nuccore/MF134659 | Publicly available at NCBI GenBank (accession no: MF134659) |
| Heintzman PD, Kapp JD, Shapiro B | 2017 | Mitochondrial genome sequence from KU 33418 | https://www.ncbi.nlm.nih.gov/nuccore/MF134660 | Publicly available at NCBI GenBank (accession no: MF134660) |
| Heintzman PD, Kapp JD, Shapiro B | 2017 | Mitochondrial genome sequence from KU 53678 | https://www.ncbi.nlm.nih.gov/nuccore/MF134661 | Publicly available at NCBI GenBank (accession no: MF134661) |
| Heintzman PD, Kapp JD, Shapiro B | 2017 | Mitochondrial genome sequence from KU 50817 | https://www.ncbi.nlm.nih.gov/nuccore/MF134662 | Publicly available at NCBI GenBank (accession no: MF134662) |
| Heintzman PD, Kapp JD, Shapiro B | 2017 | Mitochondrial genome sequence from KU 62158 | https://www.ncbi.nlm.nih.gov/nuccore/MF134663 | Publicly available at NCBI GenBank (accession no: MF134663) |
| Heintzman PD, McHorse BK, Shapiro B | 2017 | Data from: A new genus of horse from Pleistocene North America | http://dx.doi.org/10.5061/dryad.8153g | Available at Dryad Digital Repository under a CC0 Public Domain Dedication |

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

## Appendix 1

DOI: https://doi.org/10.7554/eLife.29944.019

## Supplementary methods

### Yukon sample context and identification

Pleistocene vertebrate fossils are commonly recovered at placer mining localities, in the absence of stratigraphic context, as miners are removing frozen sediments to access underlying gold bearing gravel (*Froese et al., 2009*; *Harington, 2011*). We recovered *H. francisci* fossils along with other typical late Pleistocene (Rancholabrean) taxa, including caballine horses (*Equus* sp.), woolly mammoth (*Mammuthus primigenius*), steppe bison (*Bison priscus*), and caribou (*Rangifer tarandus*), which are consistent with our age estimates based on radiocarbon dating (*Supplementary file 1*). All Yukon fossil material consisted of limb bones that were taxonomically assigned based on their slenderness and are housed in the collections of the Yukon Government (YG).

### Radiocarbon dating

We subsampled fossil specimens using handheld, rotating cutting tools and submitted them to either the KECK Accelerator Mass Spectrometry (AMS) Laboratory at the University of California (UC), Irvine (UCIAMS) and/or the Center for AMS (CAMS) at the Lawrence Livermore National Laboratory. We extracted collagen from the fossil subsamples using ultrafiltration (*Beaumont et al., 2010*), which was used for AMS radiocarbon dating. We were unable to recover collagen from TMM 34–2518 (*francisci* holotype), consistent with the probable middle Pleistocene age of this specimen (*Lundelius and Stevens, 1970*). We recovered finite radiocarbon dates from all other fossils, with the exception of the two *Equus* cf. *scotti* specimens. We calibrated AMS radiocarbon dates using the IntCal13 curve (*Reimer et al., 2013*) in OxCal v4.2 (https://c14.arch.ox.ac.uk/oxcal/OxCal.html) and report median calibrated dates in *Supplementary file 1*.

### Morphometric analysis of third metatarsals

Third metatarsal (MTIII) and other elemental measurements were either taken by GDZ or ES or from the literature (*Figure 2—source data 1*). For morphometric analysis, we focused exclusively on MTIIIs, which exhibit notable differences in slenderness among equid groups (*Figure 2—figure supplement 2a*; [*Weinstock et al., 2005*]). Starting with a data set of 10 variables (following [*Eisenmann et al., 1988*]), we compared the loadings of all variables in principal components space in order to remove redundant measurements. This reduced the data set to four variables (GL: greatest length, Pb: proximal breadth, Mb: midshaft breadth, and DABm: distal articular breadth at midline). We visualized the reduced variables using principal components analysis, computed in R (*Appendix 1—table 2—source data 1*; [*R Development Core Team, 2008*]), and performed logistic regression on the first three principal components to test whether MTIII morphology can distinguish stilt-legged (hemionine *Equus* and *H. francisci*, n = 105) from stout-legged (caballine *Equus*, n = 187) equid specimens.

### Target enrichment and sequencing

We enriched libraries for equid mitochondrial DNA following the MyBaits v2 protocol (Arbor Biosciences, Ann Arbor, MI), with RNA bait molecules constructed from the horse reference mitochondrial genome sequence (NC_001640.1). We then sequenced the enriched libraries for 2 × 150 cycles on the Illumina HiSeq-2000 platform at UC Berkeley or 2 × 75 cycles on the MiSeq platform at UC Santa Cruz, following the manufacturer's instructions. We

produced data for the nuclear genomic analyses by shotgun sequencing 17 of the unenriched libraries for 2 × 75 cycles on the MiSeq to produce ~1.1–6.4 million reads per library (*Figure 1—source data 2*).

## Mitochondrial genome reconstruction

We initially reconstructed the mitochondrial genome for *H. francisci* specimen YG 404.663 (PH047). For sequence data enriched for the mitochondrial genome, we trimmed adapter sequences, merged paired-end reads (with a minimum overlap of 15 base pairs (bp) required), and removed merged reads shorter than 25 bp, using SeqPrep (*St. John, 2013*; https://github.com/jstjohn/SeqPrep). We then mapped the merged and remaining unmerged reads to the horse reference mitochondrial genome sequence using the Burrows-Wheeler Aligner aln (BWA-aln v0.7.5; [*Li and Durbin, 2010*]), with ancient parameters (-l 1024; [*Schubert et al., 2012*]). We removed reads with a mapping quality less than 20 and collapsed duplicated reads to a single sequence using SAMtools v0.1.19 rmdup (*Li et al., 2009*). We called consensus sequences using Geneious v8.1.7 (Biomatters, http://www.geneious.com; [*Kearse et al., 2012*]). We then re-mapped the reads to the same reference mitochondrial genome using the iterative assembler, MIA (*Briggs et al., 2009*). Consensus sequences from both alignment methods required each base position to be covered a minimum of three times, with a minimum base agreement of 67%. The two consensus sequences were then combined to produce a final consensus sequence for YG 404.663 (Genbank: KT168321), which we used as the *H. francisci* reference mitochondrial genome sequence.

For the remaining newly analyzed 21 *H. francisci*, two *E.* cf. *scotti*, and two *E. lambei* samples, we merged and removed reads as described above. We then separately mapped the retained reads to the horse and *H. francisci* mitochondrial reference genome sequences using MIA. Consensus sequences from MIA analyses were called as described above. The two consensus sequences were then combined to produce a final consensus sequence for each sample, with coverage ranging from 5.8× to 110.7× (*Supplementary file 1*). We also reconstructed the mitochondrial genomes for four previously published samples: YG 401.268, LACM(CIT) 109/150807, KU 62158, and KU 62055 (*Supplementary file 1*; [*Vilstrup et al., 2013*; *Weinstock et al., 2005*]).

## Mitochondrial genome alignments

We supplemented our 30 new mitochondrial genome sequences with 38 previously published complete equid mitochondrial genomes, which included all extant *Equus* species, and extinct *Hippidion*, *E. ovodovi*, and *E.* cf. *scotti* ('equids'). We constructed six alignment data sets for the mitochondrial genome analyses: (1) equids and White rhinoceros (*Ceratotherium simum*; NC_001808) (n = 69); (2) equids and Malayan tapir (*Tapirus indicus*; NC_023838) (n = 69); (3) equids, six rhinos, two tapirs, and dog (*Canis lupus familiaris*; NC_002008) (n = 77); (4) equids, six rhinos, two tapirs, 19 published equid short fragments, and two published NWSL equid mitochondrial genome sequences (n = 88); (5) a reduced equid data set (n = 32); and (6) a full equid data set (n = 68) (*Heintzman et al., 2017*). For data sets three and four, we selected one representative from all rhino and tapir species for which full mitochondrial genome data are publicly available (*Supplementary file 1*).

For all six data sets, we first created an alignment using muscle (v3.8.31; [*Edgar, 2004*]). We then manually scrutinized alignments for errors and removed a 253 bp variable number of tandem repeats (VNTR) part of the control region, corresponding to positions 16121–16373 of the horse reference mitochondrial genome. We partitioned the alignments into six partitions (three codon positions, ribosomal-RNAs, transfer-RNAs, and control region), using the annotated horse reference mitochondrial genome in Geneious, following (*Heintzman et al., 2015*). We excluded the fast-evolving control region alignment for data set three, which included the highly-diverged dog sequence. For each partition, we selected models of molecular evolution using the Bayesian information criterion in jModelTest (v2.1.6; [*Darriba et al., 2012*]) (*Appendix 1—table 1*).

**Appendix 1—table 1.** Selected models of molecular evolution for partitions of the first five mtDNA genome alignment data sets. All lengths are in base pairs. Reduced length excludes the Coding3 and CR partitions. For all RAxML analyses the GTR model was selected, but this cannot be implemented in MrBayes and so the HKY model was used. EPA: evolutionary placement algorithm; CR: control region. *The TrN model was implemented.

| Data set | | Partition | | | | | | Total length | |
|---|---|---|---|---|---|---|---|---|---|
| | | Coding1 | Coding2 | Coding3 | rRNAs | tRNAs | CR | All | Reduced |
| 1. White rhino outgroup | Length | 3803 | 3803 | 3803 | 2579 | 1529 | 1066 | 16583 | 11714 |
| | Model | GTR + I + G | HKY + I + G | GTR + I + G | GTR + I + G | HKY + I + G | HKY*+I + G | | |
| 2. Malayan tapir outgroup | Length | 3803 | 3803 | 3803 | 2585 | 1530 | 1065 | 16589 | 11721 |
| | Model | GTR + I + G | HKY + I + G | GTR + I + G | GTR + I + G | HKY + I + G | HKY*+G | | |
| 3. Dog + ceratomorphs outgroups | Length | 3803 | 3803 | 3803 | 2615 | 1540 | N/A | 15564 | 11761 |
| | Model | GTR + I + G | HKY + I + G | GTR + I + G | GTR + I + G | HKY + I + G | N/A | | |
| 4. EPA | Length | 3803 | 3803 | 3803 | 2601 | 1534 | 1118 | 16662 | 11741 |
| | Model | GTR + I + G | TrN + I + G | GTR + I + G | GTR + I + G | HKY + I + G | HKY + I + G | | |
| 5. Equids only | Length | 3802 | 3802 | 3802 | 2571 | 1528 | 971 | 16476 | 11703 |
| | Model | TrN + I + G | TrN + I + G | GTR + G | TrN + I + G | HKY + I | HKY + G | | |

DOI: https://doi.org/10.7554/eLife.29944.020

## Phylogenetic analysis of mitochondrial genomes

To test the phylogenetic position of the NWSL equids, we conducted Bayesian and maximum likelihood (ML) phylogenetic analyses of data sets one, two, and three, under the partitioning scheme and selected models of molecular evolution described above. For outgroup, we selected: White rhinoceros (data set one), Malayan tapir (data set two), or dog (data set three). For each of the data sets, we varied the analyses based on (a) inclusion or exclusion of the fast-evolving partitions (third codon positions and control region, where appropriate) and (b) inclusion or exclusion of the *Hippidion* sequences. We ran Bayesian analyses in MrBayes (v3.2.6, [*Ronquist et al., 2012*]) for two parallel runs of 10 million generations, sampling every 1,000, with the first 25% discarded as burn-in. We conducted ML analyses in RAxML (v8.2.4, [*Stamatakis, 2014*]), using the GTRGAMMAI model across all partitions, and selected the best of three trees. We evaluated branch support with both Bayesian posterior probability scores from MrBayes and 500 ML bootstrap replicates in RAxML.

## Placement of outgroups and published sequences *a posteriori*

We used the evolutionary placement algorithm (EPA) in RAxML to determine the *a posteriori* likelihood of phylogenetic placements for eight candidate equid outgroups (two tapirs, six rhinos) relative to the four well supported major equid groups (*Hippidion*, NWSL equids, caballine *Equus*, non-caballine *Equus*). We first constructed an unrooted reference tree consisting only of the equids from data set four in RAxML. We then analyzed the placements of the eight outgroups and retaining all placements up to a cumulative likelihood threshold of 0.99. We used the same approach to assess the placement of 21 previously published equid sequences derived from 13 NWSL equids (*Barrón-Ortiz et al., 2017*; *Vilstrup et al., 2013*; *Weinstock et al., 2005*), five *Hippidion devillei* (*Orlando et al., 2009*), and three *E. ovodovi* (*Orlando et al., 2009*) (*Appendix 2—table 3*).

## Divergence date estimation from mitochondrial genomes

To further investigate the topology of the four major equid groups, and to infer divergence times between them, we ran Bayesian timetree analyses in BEAST (v1.8.4; [*Drummond et al., 2012*]). Unlike the previous analyses, BEAST can resolve branching order in the absence of an outgroup, by using branch length and molecular clock methods. For BEAST analyses, we used data set five. We did not enforce monophyly. Where available, we used radiocarbon dates to tip date ancient samples. For two samples without available radiocarbon dates, we sampled the ages of tips. For the *E. ovodovi* sample (mtDNA genome: NC_018783), which was found in a cave that has been stratigraphically dated as late Pleistocene and includes other *E. ovodovi* remains have been dated to ~45–50 ka BP (*Eisenmann and Sergej, 2011*; *Orlando et al., 2009*), we used the following lognormal prior (mean: $4.5 \times 10^4$, log(stdev): 0.766, offset: $1.17 \times 10^4$) to ensure that 95% of the prior fell within the late Pleistocene (11.7–130 ka BP). For the *E.* cf. *scotti* mitochondrial genome (KT757763), we used a normal prior (mean: $6.7 \times 10^5$, stdev: $5.64 \times 10^4$) to ensure that 95% of the prior fell within the proposed age range of this specimen (560–780 ka BP; [*Orlando et al., 2013*]). We further calibrated the tree using an age of 4–4.5 Ma for the root of crown group *Equus* (normal prior, mean $4.25 \times 10^6$, stdev: $1.5 \times 10^5$) (*Orlando et al., 2013*). To assess the impact of variables on the topology and divergence times, we either (a) included or excluded the fast-evolving partitions, (b) constrained the root height (lognormal prior: mean $1 \times 10^7$, stdev: 1.0) or not, and (c) included or excluded the *E. ovodovi* sequence, which was not directly dated. We used the models of molecular evolution estimated by jModeltest (*Appendix 1—table 1*). We estimated the substitution and clock parameters for each partition, and estimated a single tree using all partitions. We implemented the birth-death serially sampled (BDSS) tree prior. We ran two analyses for each variable combination. In each analysis, we ran the MCMC chain for 100 million generations, sampling trees and parameters every 10,000, and discarding the first 10%

as burn-in. We checked log files for convergence in Tracer (v1.6; http://tree.bio.ed.ac.uk/software/tracer/). We combined trees from the two runs for each variable combination in LogCombiner (v1.8.4) and then calculated the maximum clade credibility (MCC) tree in TreeAnnotator (v1.8.4). We report divergence dates as 95% highest posterior probability credibility intervals of node heights.

## Mitochondrial synapomorphy analysis

We first divided data set six, which consists of all available and complete equid mitogenomic sequences, into three data sets based on the genera *Hippidion*, *Haringtonhippus*, and *Equus*. For each of the three genus-specific alignments, we created a strict consensus sequence, whereby sites were only called if there was 100% sequence agreement, whilst including gaps and excluding ambiguous sites. We then compared the three genus-specific consensus sequences to determine sites where one genus exhibited a base state that is different to the other two genera, or, at five sites, where each genus has its own base state (*Appendix 1—table 2—source data 1*). In this analysis, we did not make any inference regarding the ancestral state for the identified synapomorphic base states. We identified 391 putative mtDNA genome synapomorphies for *Hippidion*, 178 for *Haringtonhippus*, and 75 for *Equus* (*Appendix 1—table 2*; *Appendix 1—table 2—source data 1*).

**Appendix 1—table 2.** Summary of the number and type of synapomorphic bases for each of the three examined equid genera. A full list of these substitutions, and their position relative to the *E. caballus* reference mitochondrial genome (NC_001640), can be found in *Appendix 1—table 2*-Source data 1. *total includes a further five synapomorphic sites that have unique states in each genus.

| Substitution | Hippidion | Haringtonhippus | Equus |
|---|---|---|---|
| Transition | 338 | 147 | 66 |
| Transversion | 43 | 22 | 4 |
| Insertion | 2 | 4 | 0 |
| Deletion | 3 | 0 | 0 |
| Total* | 391 | 178 | 75 |

DOI: https://doi.org/10.7554/eLife.29944.021

The following source data available for Appendix 1—table 2:

**Appendix 1—table 2—Source data 1.** A compilation of all 634 putative synapomorphic sites in the mitochondrial genome for *Hippidion*, *Haringtonhippus*, and *Equus* (A), with a comparison to the published MS272 mitochondrial genome sequence at the 140 sites with a base state that matches one of the three genera (B).

The horse reference mtDNA has Genbank accession NC_001640.1.

DOI: https://doi.org/10.7554/eLife.29944.022

## Phylogenetic inference from nuclear genomes

We compared the genomes of a horse (*E. caballus*; EquCab2; GCA_000002305.1) and donkey (*E. asinus*; Willy, 12.4×; http://geogenetics.ku.dk/publications/middle-pleistocene-omics; [*Orlando et al., 2013*]) with shotgun sequence data from 17 of our NWSL equid samples (*Figure 1—source data 2*, and *Supplementary file 1*). We merged paired-end reads using SeqPrep as described above, except that we removed merged reads shorter than 30 bp. We further removed merged and remaining unmerged reads that had low sequence complexity, defined as a DUST score >7, using PRINSEQ-lite v0.20.4 (*Schmieder and Edwards, 2011*). We used four successive approaches to minimize the impact of mapping bias introduced from ancient DNA fragment length variation and reference genome choice.

We first followed a modified version of the approach outlined in (*Heintzman et al., 2015*). We mapped the donkey genome to the horse genome by computationally dividing the

donkey genome into 150 bp 'pseudo-reads' tiled every 75 bp, and aligned these pseudo-reads using Bowtie2-local v2.1.0 (*Langmead and Salzberg, 2012*) while allowing one seed mismatch and a maximum mismatch penalty of four to better account for ancient DNA specific damage (*Appendix 1—figure 1*, steps 1–3). We then mapped the filtered shotgun data from each of the NWSL equid samples to the horse genome using Bowtie2-local with the settings described above, and removed PCR duplicated reads and those with a mapping quality score of <30 in SAMtools. We called a pseudo-haploidized sequence for the donkey and NWSL equid alignments, by randomly picking a base with a base quality score $\geq$60 at each position, using SAMtools mpileup. We masked positions that had a coverage not equal to 2$\times$ (donkey) or 1$\times$ (NWSL equid), and those located on scaffolds shorter than 100 kb (*Appendix 1—figure 1*, step 4). As the horse, donkey, and NWSL equid genome sequences were all based on the horse genome coordinates, we compared the relative transversion frequency between the donkey or NWSL equids and the horse using custom scripts. We restricted our analyses to transversions to avoid the impacts of ancient DNA damage, which can manifest as erroneous transitions from the deamination of cytosine (e.g. *Appendix 2—figure 1,2*) (*Dabney et al., 2013b*). We repeated this analysis, but with the horse and NWSL equids mapped to the donkey genome (the donkey genome coordinate framework).

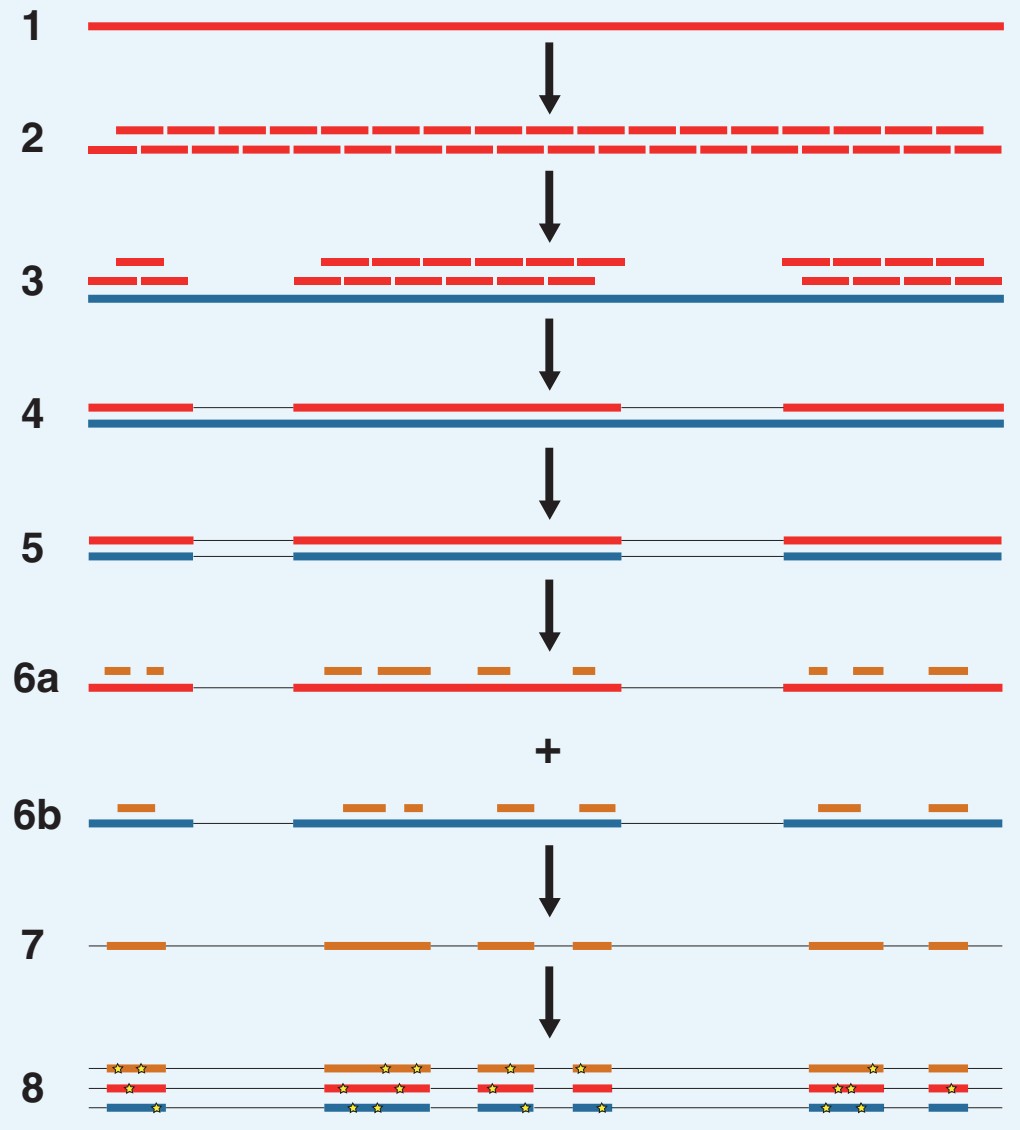

**Appendix 1—figure 1.** An overview of the nuclear genome analysis pipeline. A first reference genome sequence (red; step 1) is divided into 150 bp pseudo-reads, tiled every 75 bp for

exactly 2 × genomic coverage (step 2). These pseudo-reads are then mapped to a second reference genome (blue; step 3), and a consensus sequence of the mapped pseudo-reads is called (step 4). Regions of the second reference genome that are not covered by the pseudo-reads are masked (step 5). For each NWSL equid sample, reads (orange) are mapped independently to the first reference consensus sequence (step 6a) and masked second reference genome (step 6b). Alignments from steps 6a and 6b are then merged (step 7). For alignment coordinates that have base calls for the first reference, second reference, and NWSL equid sample genomes, the relative frequencies of private transversion substitutions (yellow stars) for each genome are calculated (step 8). The co-ordinates from the second reference genome (blue) are used for each analysis.
DOI: https://doi.org/10.7554/eLife.29944.023

For the second approach and using the horse genome coordinate framework, we next masked sites in the horse reference genome that were not covered by donkey reads at a depth of 2×. This resulted in the horse genome and donkey consensus sequence being masked at the same positions (*Appendix 1—figure 1*, step 5). We then separately mapped the filtered NWSL equid shotgun data to scaffolds longer than 100 kb for the masked horse genome and donkey consensus sequence (*Appendix 1—figure 1*, step 6), called NWSL equid consensus sequences, and calculated relative transversion frequencies as described above. This analysis was repeated using the donkey genome coordinate framework.

Next, for each genome coordinate framework, we combined the two alignments for each NWSL equid sample from approach two to create a union of reads mappable to both the masked coordinate genome and alternate genome consensus sequence (*Appendix 1—figure 1*, step 7). If a NWSL equid read mapped to different coordinates between the two references, we selected the alignment with the higher map quality score and randomly selected between mappings of equal quality. We then called NWSL equid consensus sequences as above. As this third approach allowed for simultaneous comparison of the horse, donkey, and NWSL equid sequences, we calculated relative private transversion frequencies for each sequence, at sites where all three sequences had a base call, using tri-aln-report (*Green et al., 2015*); https://github.com/Paleogenomics/Chrom-Compare) (*Appendix 1—figure 1*, step 8).

Finally, as a fourth approach and for both genome coordinate frameworks, we repeated approach three with the exception that we divided the NWSL alignments by mapped read length. We split the alignments into 10 bp read bins ranging from 30–39 to 120–129 bp, and discarded longer reads and paired-end reads that were unmerged by SeqPrep. We called consensus sequences and calculated relative private transversion frequencies for each sequence as described above. We only used relative private transversion frequencies from the 90–99 to 120–129 bp bins for divergence date estimates (Appendix 2).

## Sex determination from nuclear genomes

We used the alignments of the 17 NWSL equids to the horse genome, from approach one described above, to infer the probable sex of these individuals. For this, we determined the number of reads mapped to each chromosome using SAMtools idxstats. For each chromosome, we then calculated the relative mapping frequency by dividing the number of mapped reads by the length of the chromosome. We then compared the relative mapping frequency between the autosomes and X-chromosome. As males and females are expected to have one and two copies of the X chromosome, respectively, and two copies of every autosome, we inferred a male if the ratio between the autosomes and X-chromosome was 0.45–0.55 and a female if the ratio were 0.9–1.1.

## DNA damage analysis

For a subset of nine samples, we realigned the filtered sequence data from the libraries enriched for equid mitochondrial DNA to either the *H. francisci* (for *H. francisci* samples) or horse (for *E. lambei* and *E.* cf. *scotti* samples) reference mitochondrial genome sequences

using BWA-aln as described above. We also realigned the filtered unenriched sequence data to the horse reference genome (EquCab2) for a subset of six samples using the same approach. We then analyzed patterns of DNA damage in mapDamage v2.0.5 (*Jónsson et al., 2013*).

## Appendix 2

DOI: https://doi.org/10.7554/eLife.29944.024

## Supplementary Results

### Ancient DNA characterization

We selected a subset of samples for the analysis of DNA damage patterns. In all of these samples, we observe expected patterns of damage in both mitochondrial and nuclear DNA, including evidence of the deamination of cytosine residues at the ends of reads, depurination-induced strand breaks, and a short mean DNA fragment length (*Dabney et al., 2013b*) (*Appendix 2—figure 1–2*). We note that the sample with the greatest proportion of deaminated cytosines is *E*. cf. *scotti* (YG 198.1; *Appendix 2—figure 1v-x*), which is the oldest sample in the subset (*Supplementary file 1*).

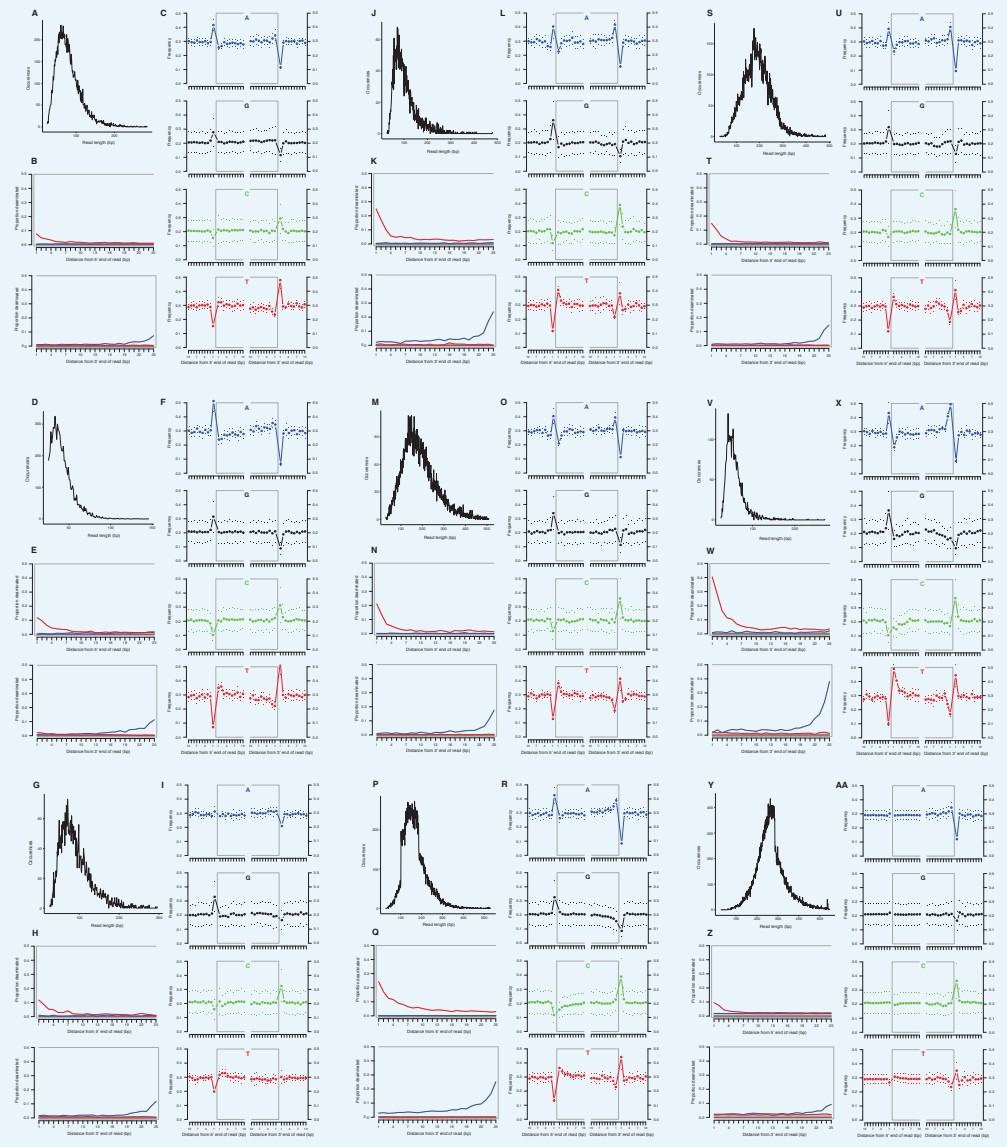

**Appendix 2—figure 1.** Characterization of ancient mitochondrial DNA damage patterns from nine equid samples. *H. francisci*: (**A–C**) JK166 (LACM(CIT) 109/150807; Nevada), (**D–F**) JK207 (LACM(CIT) 109/156450; Nevada), (**G–I**) JK260 (KU 47800; Wyoming), (**J–L**) PH013 (YG 130.6;

Yukon), (**M–O**) PH047 (YG 404.663; Yukon), (**P–R**) MS272 (YG 401.268; Yukon), (**S–U**) MS349 (YG 130.55; Yukon); *E.* cf. *scotti*: (**V–X**) PH055 (YG 198.1; Yukon); *E. lambei*: (**Y–AA**) MS316 (YG 328.54; Yukon). Every third panel: (**A**) to (**Y**) DNA fragment length distributions; (**B**) to (**Z**) proportion of cytosines that are deaminated at fragment ends (red: cytosine → thymine; blue: guanine → adenine); and (**C**) to (**AA**) mean base frequencies immediately upstream and downstream of the 5' and 3' ends of mapped reads.

DOI: https://doi.org/10.7554/eLife.29944.025

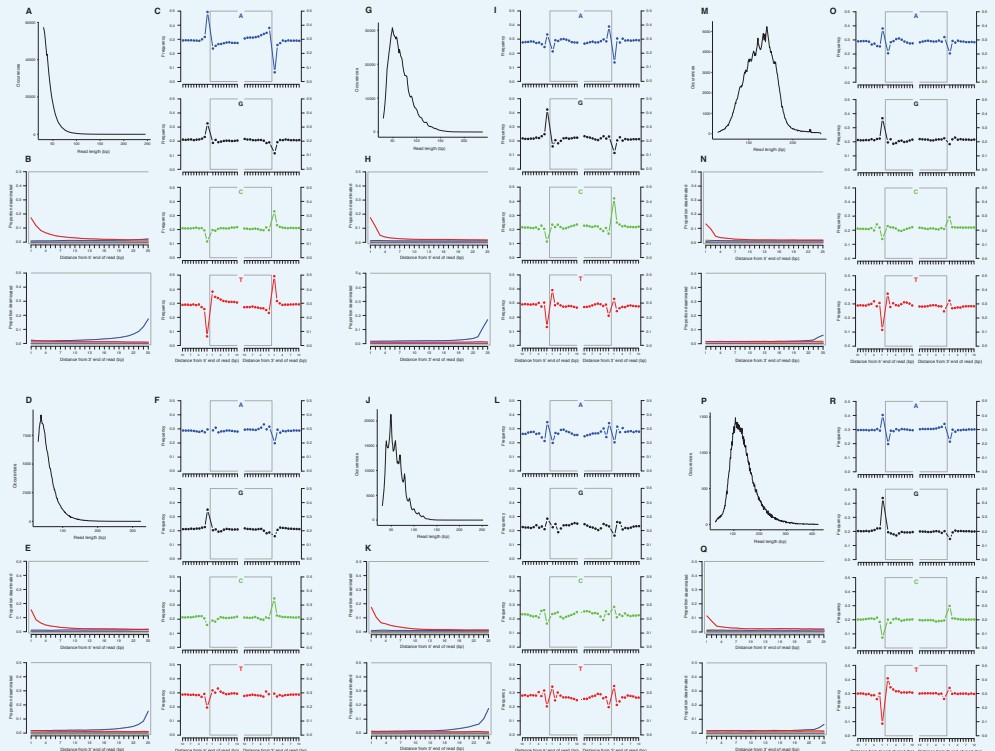

**Appendix 2—figure 2.** Characterization of ancient nuclear DNA damage patterns from six *H. francisci* samples. (**A–C**) JK166 (LACM(CIT) 109/150807; Nevada), (**D–F**) JK260 (KU 47800; Wyoming), (**G–I**) PH013 (YG 130.6; Yukon), (**J–L**) PH036 (YG 76.2; Yukon), (**M–O**) MS349 (YG 130.55; Yukon), (**P–R**) MS439 (YG 401.387; Yukon). Every third panel: (**A**) to (**P**) DNA fragment length distributions; (**B**) to (**Q**) proportion of cytosines that are deaminated at fragment ends (red: cytosine → thymine; blue: guanine → adenine); and (**C**) to (**R**) mean base frequencies immediately upstream and downstream of the 5' and 3' ends of mapped reads.

DOI: https://doi.org/10.7554/eLife.29944.026

## Resolving the phylogenetic placement of NWSL equids using mitochondrial genomes

We ran Bayesian and ML phylogenetic analyses on mtDNA genome alignment data sets 1–3, whilst varying the outgroup, including (all) or excluding (reduced) the fast-evolving partitions (see Appendix 1), and including or excluding the *Hippidion* sequences. In all analyses, we recover four major equid groups (*Hippidion*, NWSL equids(=*H. francisci*), caballine *Equus*, and non-caballine *Equus*) with strong statistical support (Bayesian posterior probability (BPP): 1.000; ML bootstrap: 96–100%; *Appendix 2—table 1*), consistent with previous studies (e.g. [*Der Sarkissian et al., 2015*; *Orlando et al., 2009*]). We recover conflicting phylogenetic topologies between these four groups, however, which is dependent on the variables described above and the choice of phylogenetic algorithm (*Appendix 2—figure 3*; *Appendix 2—table 1*). Across all analyses, strong statistical support (BPP:≥0.99; ML bootstrap:≥95%) is only associated with topology 1 (*Appendix 2—figure 3*; *Appendix 2—table 1*), in which NWSL equids are placed outside of *Equus*, and *Hippidion* is placed outside

**Appendix 2—table 1.** Topological shape and support values for the best supported trees. These results are from the Bayesian and maximum likelihood (ML) analyses of mtDNA data sets 1–3, including either the all or reduced partition sets, and with *Hippidion* sequences either included or excluded. Topology numbers and node letters refer to those outlined in *Appendix 2—figure 3*. Bayesian posterior probability support of >0.99 and ML bootstrap support of >95% are in bold for nodes A and B. *support for nodes that are consistent with topology one in *Appendix 2—figure 3*. NCs: non-caballines.

| Outgroup | Partitions | Hippidion? | Tips | Analysis method | Topology | Support | | Hippidion | NWSL | NCs | Caballines |
|---|---|---|---|---|---|---|---|---|---|---|---|
| | | | | | | Node A | Node B | | | | |
| White rhino (Data set 1) | All | Excluded | 63 | Bayesian | 1/2/3 | **0.996*** | N/A | N/A | 1.000 | 1.000 | 1.000 |
| | | | | ML | 1/2/3 | 71* | N/A | N/A | 100 | 99 | 100 |
| | | Included | 69 | Bayesian | 2 | 0.751 | **1.000*** | 1.000 | 1.000 | 1.000 | 1.000 |
| | | | | ML | 1 | 64* | **96*** | 100 | 100 | 100 | 100 |
| | Reduced | Excluded | 63 | Bayesian | 1/2/3 | **1.000*** | N/A | N/A | 1.000 | 1.000 | 1.000 |
| | | | | ML | 1/2/3 | **100*** | N/A | N/A | 99 | 100 | 100 |
| | | Included | 69 | Bayesian | 2 | 0.948 | **1.000*** | 1.000 | 1.000 | 1.000 | 1.000 |
| | | | | ML | 2 | 73 | **98*** | 100 | 99 | 100 | 100 |
| Malayan tapir (Data set 2) | All | Excluded | 63 | Bayesian | 5/7 | 0.971 | N/A | N/A | 1.000 | 1.000 | 1.000 |
| | | | | ML | 5/7 | 87 | N/A | N/A | 100 | 99 | 99 |
| | | Included | 69 | Bayesian | 6 | 0.808 | 0.867 | 1.000 | 1.000 | 1.000 | 1.000 |
| | | | | ML | 6 | 55 | 63 | 100 | 100 | 100 | 100 |
| | Reduced | Excluded | 63 | Bayesian | 1/2/3 | 0.675* | N/A | N/A | 1.000 | 1.000 | 1.000 |
| | | | | ML | 4/6 | 28 | N/A | N/A | 100 | 96 | 98 |
| | | Included | 69 | Bayesian | 3 | 0.685 | 0.864* | 1.000 | 1.000 | 1.000 | 1.000 |
| | | | | ML | 3 | 70 | 69 | 100 | 100 | 100 | 100 |

*Appendix 2—table 1 continued on next page*

*Appendix 2—table 1 continued*

| Outgroup | Partitions | Hippidion? | Tips | Analysis method | Topology | Support | | Hippidion | NWSL | NCs | Caballines |
| | | | | | | Node A | Node B | | | | |
| --- | --- | --- | --- | --- | --- | --- | --- | --- | --- | --- | --- |
| Dog + ceratomorphs (Data set 3) | All | Excluded | 71 | Bayesian | 1/2/3 | 0.598* | N/A | N/A | 1.000 | 1.000 | 1.000 |
| | | | | ML | 4/6 | 59 | N/A | N/A | 100 | 100 | 100 |
| | | Included | 77 | Bayesian | 1 | 1.000* | 1.000* | 1.000 | 1.000 | 1.000 | 1.000 |
| | | | | ML | 1 | 94* | 96* | 100 | 100 | 100 | 100 |
| | Reduced | Excluded | 71 | Bayesian | 1/2/3 | 0.999* | N/A | N/A | 1.000 | 1.000 | 1.000 |
| | | | | ML | 1/2/3 | 97* | N/A | N/A | 100 | 100 | 100 |
| | | Included | 77 | Bayesian | 1 | 1.000* | 1.000* | 1.000 | 1.000 | 1.000 | 1.000 |
| | | | | ML | 1 | 99* | 100* | 100 | 100 | 100 | 100 |

DOI: https://doi.org/10.7554/eLife.29944.028

of the NWSL equid-*Equus* clade. We note that the analyses with the strongest support consist of multiple outgroups (mtDNA data set three).

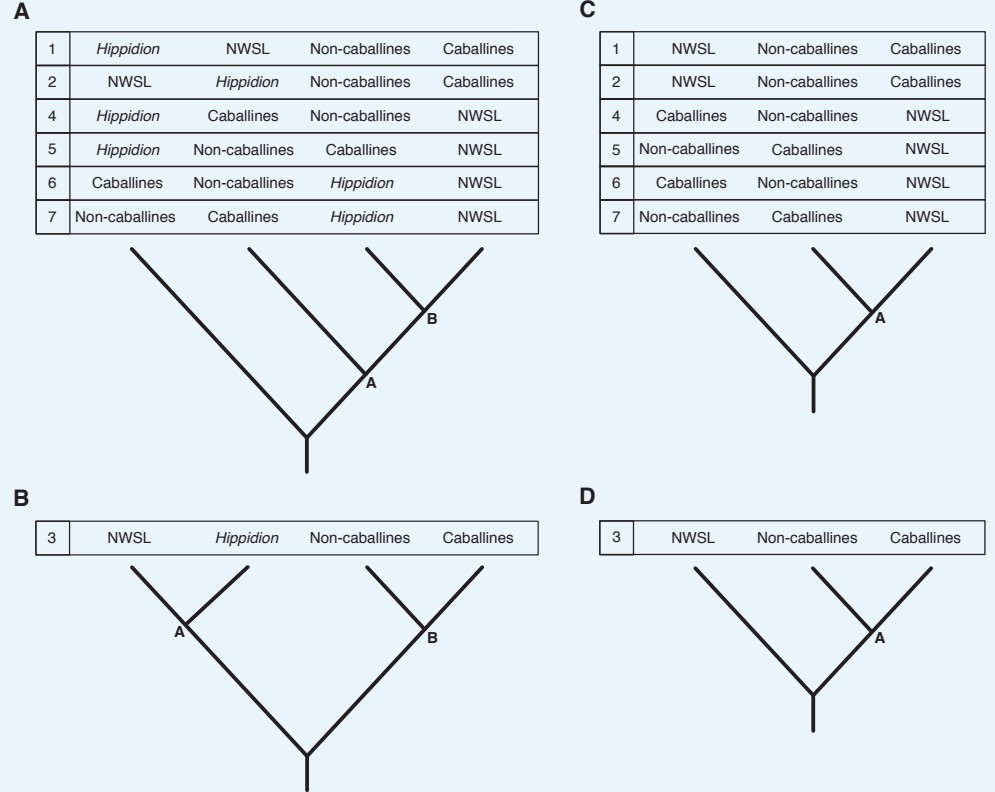

**Appendix 2—figure 3.** Seven phylogenetic hypotheses for the four major groups of equids with sequenced mitochondrial genomes. These major groups are *Hippidion*, the New World stilt-legged equids (=*Haringtonhippus*), non-caballine *Equus* (asses, zebras, and *E. ovodovi*) and caballine *Equus* (horses). (**A**) imbalanced and (**B**) balanced hypotheses. The hypotheses presented in (**C**) and (**D**) are identical to (**A**) and (**B**), except that *Hippidion* is excluded. Node letters are referenced in *Appendix 2—tables 1–2*. We only list combinations that were recovered by our palaeogenomic, or previous palaeogenetic, analyses.

**Appendix 2—table 2.** The *a posteriori* phylogenetic placement likelihood for eight ceratomorph (rhino and tapir) outgroups. These analyses used a ML evolutionary placement algorithm, whilst varying the partition set used (all or reduced), and either including or excluding *Hippidion* sequences. Likelihoods >0.95 are in bold. Topology numbers refer to those outlined in *Appendix 2—figure 3*. Genbank accession numbers are given in parentheses after outgroup names.

| | | *Hippidion?* | Included | | | | Excluded | | |
|---|---|---|---|---|---|---|---|---|---|
| **Partitions** | **Outgroup** | **Topology** | **1** | **2** | **3** | **6** | **1/2/3** | **4/6** | **5/7** |

*Appendix 2—table 2 continued on next page*

Appendix 2—table 2 continued

| Partitions | Outgroup | *Hippidion?* Topology | Included 1 | 2 | 3 | 6 | Excluded 1/2/3 | 4/6 | 5/7 |
|---|---|---|---|---|---|---|---|---|---|
| All | *Tapirus terrestris* (AJ428947) | | 0.456 | 0.317 | 0.205 | 0.018 | 0.549 | 0.313 | 0.139 |
| | *Tapirus indicus* (NC023838) | | 0.275 | 0.105 | 0.225 | 0.389 | 0.050 | 0.908 | 0.042 |
| | *Coelodonta antiquitatis* (NC012681) | | 0.998 | | | | 0.248 | 0.451 | 0.301 |
| | *Dicerorhinus sumatrensis* (NC012684) | | 0.981 | | 0.009 | | 0.155 | 0.553 | 0.292 |
| | *Rhinoceros unicornis* (NC001779) | | 0.998 | | | | 0.529 | 0.334 | 0.137 |
| | *Rhinoceros sondaicus* (NC012683) | | 0.989 | 0.006 | | | 0.732 | 0.196 | 0.072 |
| | *Ceratotherium simum* (NC001808) | | 0.448 | 0.499 | 0.053 | | 0.949 | 0.018 | 0.033 |
| | *Diceros bicornis* (NC012682) | | 0.917 | 0.065 | 0.018 | | 0.851 | 0.073 | 0.076 |
| Reduced | *Tapirus terrestris* (AJ428947) | | 0.410 | 0.391 | 0.199 | | 0.987 | | 0.012 |
| | *Tapirus indicus* (NC023838) | | 0.536 | 0.298 | 0.166 | | 0.995 | | |
| | *Coelodonta antiquitatis* (NC012681) | | 0.411 | 0.554 | 0.035 | | 1.000 | | |
| | *Dicerorhinus sumatrensis* (NC012684) | | 0.983 | 0.015 | | | 1.000 | | |
| | *Rhinoceros unicornis* (NC001779) | | 0.998 | | | | 1.000 | | |
| | *Rhinoceros sondaicus* (NC012683) | | 0.895 | 0.102 | | | 1.000 | | |
| | *Ceratotherium simum* (NC001808) | | 0.296 | 0.704 | | | 1.000 | | |
| | *Diceros bicornis* (NC012682) | | 0.996 | | | | 1.000 | | |

DOI: https://doi.org/10.7554/eLife.29944.029

We further investigated the effect of outgroup choice by using an evolutionary placement algorithm (EPA; [*Berger et al., 2011*]) to place the outgroup sequences into an unrooted ML phylogeny *a posteriori* using the same set of variables described above. We find that the outgroup placement likelihood is increased with the inclusion of *Hippidion* sequences, and that the only placements with a likelihood of ≥0.95 are consistent with topology one (*Appendix 2—figure 3*; *Appendix 2—table 2*), in agreement with the Bayesian and ML phylogenetic analyses. The phylogenetic and EPA analyses demonstrate that outgroup choice can greatly impact equid phylogenetic inference and that multiple outgroups should be used for resolving relationships between major equid groups.

We lastly ran Bayesian timetree analyses in BEAST in the absence of an outgroup, whilst including or excluding the fast-evolving partitions, including or excluding the *E. ovodovi* sequence, and constraining the root prior or not. All BEAST analyses yielded a maximum clade credibility tree that is consistent with topology one (*Figure 1* and *Appendix 2—figure 3*) with Bayesian posterior probability support for the NWSL equid-*Equus* and *Equus* clades of 0.996–1.000 (*Figure 1—source data 1*). Altogether, the phylogenetic, EPA, and timetree analyses support topology one (*Appendix 2—figure 3*), with NWSL equids falling outside of *Equus*, and therefore the NWSL equids as a separate genus, *Haringtonhippus*.

## Placement of previously published NWSL equid sequences

To confirm that all 15 previously published NWSL equid samples with available mtDNA sequence data (*Barrón-Ortiz et al., 2017*; *Vilstrup et al., 2013*; *Weinstock et al., 2005*) belong to *H. francisci*, we either reconstructed mitochondrial genomes for these samples (JW277, JW161; [*Weinstock et al., 2005*]), placed the sequences into a ML phylogeny *a posteriori* using the EPA whilst varying the partitioning scheme and inclusion or exclusion of *Hippidion* (*Appendix 2—table 3*), or both. For JW277 and JW161, the mitochondrial genomes were consistent with those derived from the newly analyzed samples (*Figure 1—*

*figure supplement 1*). For eight other NWSL equid mitochondrial sequences (JW125, JW126, JW328, EQ3, EQ9, EQ13, EQ22, EQ41; [*Barrón-Ortiz et al., 2017*; *Vilstrup et al., 2013*; *Weinstock et al., 2005*]), including samples from Mineral Hill Cave and Dry Cave (*Supplementary file 1*), the EPA strongly supported a ML placement within the NWSL equid clade (cumulative likelihood of 0.974–1.000). The EPA placed four sequences from Dry Cave, San Josecito Cave, and the Edmonton area (EQ1, EQ4, EQ16, EQ30; [*Barrón-Ortiz et al., 2017*]) within the NWSL equid clade albeit with lower support (cumulative likelihood of 0.703–0.854). We note that in the case of EQ4 from Edmonton, this may be due to very limited available sequence data (117 bp). For EQ1, EQ16, and EQ30, the placement with the second greatest support is the branch leading to NWSL equids (cumulative likelihood of 0.138–0.259), which, assuming high fidelity of the sequence data, may indicate that these samples fall outside of, but close to, sampled NWSL equid mitochondrial diversity. However, the EPA placed the remaining sample (MS272; [*Vilstrup et al., 2013*]) on the branch leading to NWSL equids with strong support (likelihood: 1.000). We therefore explored whether this is real or if the published sequence for MS272 was problematic.

We first tested the EPA on eight other equid mitochondrial sequences (*E. ovodovi*, n = 3; *Hippidion devillei*, n = 5), which grouped as expected from previous analyses (likelihood: 0.999–1.000; *Appendix 2—table 3*; [*Orlando et al., 2009*]). We then used our mitochondrial genome assembly pipeline to reconstruct a consensus for MS272 from the raw data used by *Vilstrup et al. (2013)*, which resulted in a different sequence that was consistent with other NWSL equids. To confirm this new sequence, we used the original MS272 DNA extract for library preparation, target enrichment, and sequencing. The consensus from this analysis was identical to our new sequence.

We sought to understand the origins of the problems associated with the published MS272 sequence. We first applied our synapomorphy analysis. For the called bases, we found that the published MS272 sequence contained 0/384 diagnostic bases for *Hippidion*, 124/164 for *Haringtonhippus*, and 16/70 for *Equus* (*Appendix 1—table 2—source data 1*). We infer from this analysis that the published MS272 sequence is therefore ~76% *Haringtonhippus* and that ~23% originates from *Equus*. The presence of *Equus* synapomorphies could be explained by the fact that the enriched library for MS272 was sequenced on the same run as ancient caballine horses (*Equus*), thereby potentially introducing contaminating reads from barcode bleeding (*Kircher et al., 2012*), which may have been exacerbated by alignment to the modern horse reference mitochondrial genome with BWA-aln and consensus calling using SAMtools (*Vilstrup et al., 2013*). The presence of caballine horse sequence in the published MS272 mtDNA genome explains why previous phylogenetic analyses of mitochondrial genomes have recovered NWSL equids as sister to caballine *Equus* with strong statistical support (*Der Sarkissian et al., 2015*; *Vilstrup et al., 2013*).

## Resolving the phylogenetic placement of NWSL equids using nuclear genomes

The horse and donkey genomes are representative of total *Equus* genomic diversity (*Jónsson et al., 2014*), and so, if NWSL equids are *Equus*, we should expect their genomes to be more similar to either horse or donkey than to the alternative.

Initial analyses based on approach one (see Appendix 1) were inconclusive, with some NWSL equid samples appearing to fall outside of *Equus* (higher relative transversion frequency between the NWSL equid and the horse or donkey than between the horse and donkey) and others inconsistently placed in the phylogeny, appearing most closely related to horse when aligned to the horse genome and most closely related donkey when aligned to the donkey genome (*Figure 1—source data 2*). We then used approaches two and three in an attempt to standardize between the horse and donkey reference genomes, and therefore reduce potential bias introduced from the reference genome. In the latter union-based approach, mapping should not be disproportionately sensitive to regions of the genome where NWSL equids are more horse- or donkey-like. These approaches, however, were not

**Appendix 2—table 3.** The *a posteriori* phylogenetic placement likelihood for 21 published equid mitochondrial sequences. These analyses used the ML evolutionary placement algorithm, whilst varying the partition set used (all or reduced), and either including or excluding *Hippidion* sequences. Sample names are given in parentheses after the species or group name. Localities are given for NWSL equids only. Likelihoods >0.95 are in bold. *Equus* includes only caballines and non-caballine equids (NCE). **For EQ04 from Alberta, other placement likelihood values for the *Hippidion* included/excluded partitions were: Within caballines: 0.003/0.002, Sister to caballines: 0.002/0.002, Within NCE: 0.246/0.245, Sister to NCE: 0.004/0.003. No placements were returned for 'within *Hippidion*'. bp: base pairs.

| Hippidion? | Partition | Published sample | Locality | Sequence length (bp) | Placement | | | | | |
|---|---|---|---|---|---|---|---|---|---|---|
| | | | | | Sister to *E. ovodovi* | Sister to *Hippidion* | Within NWSL | Sister to NWSL | Sister to *Equus** | Other** |
| Included | All | E. ovodovi (ACAD2305) | | 688 | **1.000** | | | | | |
| | | E. ovodovi (ACAD2302) | | 688 | **1.000** | | | | | |
| | | E. ovodovi (ACAD2303) | | 688 | **1.000** | | | | | |
| | | H. devillei (ACAD3615) | | 476 | N/A | **1.000** | | | | |
| | | H. devillei (ACAD3625) | | 543 | N/A | **1.000** | | | | |
| | | H. devillei (ACAD3627) | | 543 | N/A | **1.000** | | | | |
| | | H. devillei (ACAD3628) | | 543 | N/A | **0.999** | | | | |
| | | H. devillei (ACAD3629) | | 476 | N/A | **0.999** | | | | |
| | | NWSL equid (JW125) | Klondike, YT | 720 | N/A | | **0.996** | | | |
| | | NWSL equid (JW126) | Klondike, YT | 720 | N/A | | **0.999** | | | |

*Appendix 2—table 3 continued on next page*

Appendix 2—table 3 continued

| Hippidion? | Partition | Published sample | Sequence length (bp) | Locality | Placement | | | | | |
|---|---|---|---|---|---|---|---|---|---|---|
| | | | | | Sister to E. ovodovi | Sister to Hippidion | Within NWSL | Sister to NWSL | Sister to Equus* | Other** |
| Included | All | NWSL equid (EQ01) | 620 | Dry Cave, NM | N/A | | 0.735 | 0.256 | | |
| | | NWSL equid (EQ03) | 117 | Dry Cave, NM | N/A | 0.002 | 0.974 | 0.011 | 0.003 | |
| | | NWSL equid (EQ04) | 117 | Edmonton, AB | N/A | 0.004 | 0.703 | 0.014 | 0.007 | 0.255 |
| | | NWSL equid (EQ09) | 620 | Natural Trap Cave, WY | N/A | | 0.981 | 0.014 | | |
| | | NWSL equid (EQ13) | 620 | Natural Trap Cave, WY | N/A | | 0.992 | | | |
| | | NWSL equid (EQ16) | 464 | Dry Cave, NM | N/A | | 0.854 | 0.138 | | |
| | | NWSL equid (EQ22) | 620 | Natural Trap Cave, WY | N/A | | 0.999 | | | |
| | | NWSL equid (EQ30) | 393 | San Josecito Cave, MX-NL | N/A | | 0.792 | 0.198 | | |
| | | NWSL equid (EQ41) | 398 | Natural Trap Cave, WY | N/A | | 0.997 | | | |
| | | NWSL equid (JW328) | mitogenome | Mineral Hill Cave, NV | N/A | | 1.000 | | | |
| | | NWSL equid (MS272) | mitogenome | Klondike, YT | N/A | | | 1.000 | | |
| | Reduced | NWSL equid (JW328) | mitogenome | Mineral Hill Cave, NV | N/A | | 0.996 | | | |
| | | NWSL equid (MS272) | mitogenome | Klondike, YT | N/A | | | 1.000 | | |

Appendix 2—table 3 continued on next page

Appendix 2—table 3 continued

| | | | | | Placement | | | | | |
|---|---|---|---|---|---|---|---|---|---|---|
| Hippidion? | Partition | Published sample | Sequence length (bp) | Locality | Sister to E. ovodovi | Sister to Hippidion | Within NWSL | Sister to NWSL | Sister to Equus* | Other** |
| Excluded | All | E. ovodovi (ACAD2305) | 688 | | 1.000 | N/A | | | N/A | |
| | | E. ovodovi (ACAD2302) | 688 | | 1.000 | N/A | | | N/A | |
| | | E. ovodovi (ACAD2303) | 688 | | 1.000 | N/A | | | N/A | |
| | | NWSL equid (JW125) | 720 | Klondike, YT | N/A | N/A | 0.996 | | N/A | |
| | | NWSL equid (JW126) | 720 | Klondike, YT | N/A | N/A | 0.999 | | N/A | |
| | | NWSL equid (EQ01) | 620 | Dry Cave, NM | N/A | N/A | 0.731 | 0.259 | N/A | |
| | | NWSL equid (EQ03) | 117 | Dry Cave, NM | N/A | N/A | 0.980 | 0.010 | N/A | |
| | | NWSL equid (EQ04) | 117 | Edmonton, AB | N/A | N/A | 0.721 | 0.013 | N/A | 0.252 |
| | | NWSL equid (EQ09) | 620 | Natural Trap Cave, WY | N/A | N/A | 0.987 | 0.008 | N/A | |
| | | NWSL equid (EQ13) | 620 | Natural Trap Cave, WY | N/A | N/A | 0.993 | | N/A | |
| | | NWSL equid (EQ16) | 464 | Dry Cave, NM | N/A | N/A | 0.844 | 0.148 | N/A | |
| | | NWSL equid (EQ22) | 620 | Natural Trap Cave, WY | N/A | N/A | 0.999 | | N/A | |
| | | NWSL equid (EQ30) | 393 | San Josecito Cave, MX-NL | N/A | N/A | 0.788 | 0.203 | N/A | |
| | | NWSL equid (EQ41) | 398 | Natural Trap Cave, WY | N/A | N/A | 0.995 | | N/A | |
| | | NWSL equid (JW328) | mitogenome | Mineral Hill Cave, NV | N/A | N/A | 1.000 | | N/A | |
| | | NWSL equid (MS272) | mitogenome | Klondike, YT | N/A | N/A | | 1.000 | N/A | |
| | Reduced | NWSL equid (JW328) | mitogenome | Mineral Hill Cave, NV | N/A | N/A | 0.995 | | N/A | |
| | | NWSL equid (MS272) | mitogenome | Klondike, YT | N/A | N/A | 0.995 | 1.000 | N/A | |

DOI: https://doi.org/10.7554/eLife.29944.030

successful, but we noted that relative private transversion frequency for the coordinate genome and NWSL equid sequences correlated with mean DNA fragment length (*Appendix 2—figure 4* and *Figure 1—source data 2*). We therefore used approach four to control for the large variation in mean DNA fragment length between NWSL equid sequences (*Appendix 2—figure 2* and *Figure 1—source data 2*), which is likely due to a combination of DNA preservation and differences in the DNA extraction and library preparation techniques used (*Figure 1—source data 2*). This allowed for direct comparison between the NWSL equid samples, which showed a consistent pattern across read length bins (*Figure 1—figure supplement 2*, *Figure 1—source data 1*). The relative private transversion frequency for both the coordinate genome and NWSL equid sequences increase with read length until the 90–99 bp bin, at which point the coordinate genome and alternate sequence relative private transversion frequencies converge (defined as a ratio between 0.95–1.05) and the NWSL equid relative private transversion frequencies reach plateau at between 1.40–1.56× greater than that of the horse or donkey (*Figure 1—figure supplements 2–3*, *Figure 1—source data 1*).

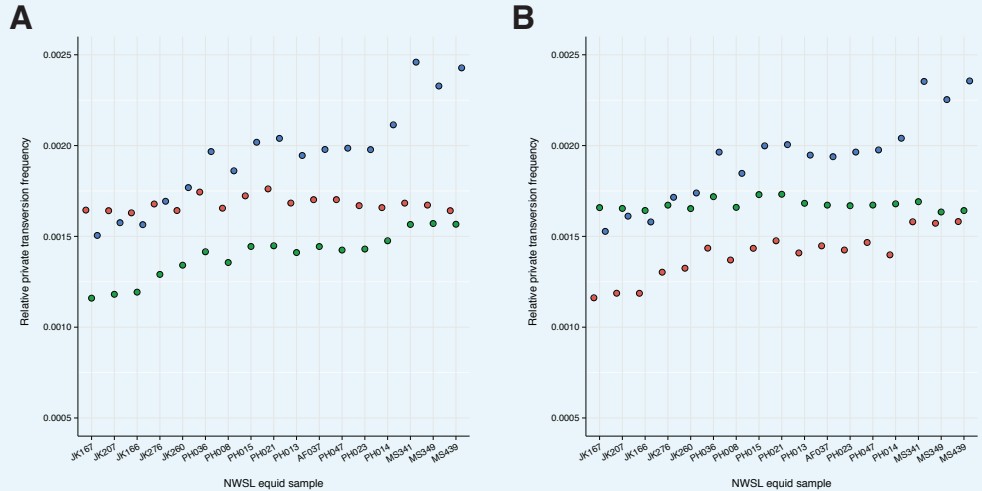

**Appendix 2—figure 4.** A comparison of relative private transversion frequencies between the nuclear genomes of a caballine *Equus* (horse, *E. caballus*; green), a non-caballine *Equus* (donkey, *E. asinus*; red), and the 17 New World-stilt legged (NWSL) equid samples (=*Haringtonhippus francisci*; blue), using approach three (Appendix 1), with samples ordered by increasing mean mapped read length. Analyses are based on alignment to the horse (**A**) or donkey (**B**) genome coordinates.
DOI: https://doi.org/10.7554/eLife.29944.031

A greater relative private transversion frequency in NWSL equids, as compared to horse and donkey, is consistent with their being more diverged than the horse-donkey split (*Equus*) and therefore supports the hypothesis of NWSL equids as a separate genus (*Haringtonhippus*).

## Sex determination from nuclear genomes

We inferred the sex of our 17 NWSL equid samples by calculating the ratio of relative mapping frequencies between the autosomes and X-chromosome (*Appendix 2—table 4—source data 1*). We find that at least four of our samples are male and at least eight are female (*Appendix 2—table 4*).

**Appendix 2—table 4.** Sex determination analysis of 17 NWSL equids. Chromosome ratio is the relative mapping frequency ratio between all autosomes and the X-chromosome. Males are inferred if the ratio is 0.45–0.55 and females if the ratio is 0.9–1.1.

| Sample | Museum accession | Chromosome ratio | Inferred sex |
|--------|-----------------|------------------|--------------|
| AF037 | YG 402.235 | 0.48 | male |
| JK166 | LACM(CIT) 109/150807 | 0.93 | female |
| JK167 | LACM(CIT) 109/149291 | 0.91 | female |
| JK207 | LACM(CIT) 109/156450 | 0.92 | female |
| JK260 | KU 47800 | 0.95 | female |
| JK276 | KU 53678 | 0.91 | female |
| MS341 | YG 303.1085 | 0.50 | male |
| MS349 | YG 130.55 | 0.48 | male |
| MS439 | YG 401.387 | 0.98 | female |
| PH008 | YG 404.205 | 0.90 | female |
| PH013 | YG 130.6 | 0.87 | probable female |
| PH014 | YG 303.371 | 0.46 | male |
| PH015 | YG 404.662 | 0.44 | probable male |
| PH021 | YG 29.169 | 0.83 | probable female |
| PH023 | YG 160.8 | 0.91 | female |
| PH036 | YG 76.2 | 0.81 | probable female |
| PH047 | YG 404.663 | 0.88 | probable female |

DOI: https://doi.org/10.7554/eLife.29944.032

The following source data available for Appendix 2—table 4:
Appendix 2—table 4—Source data 1. Data from the sex determination analyses of 17 NWSL equids, based on alignment to the horse genome (EquCab2).
DOI: https://doi.org/10.7554/eLife.29944.033

We note that all three Gypsum Cave samples are inferred to be female, have statistically indistinguishable radiocarbon dates, and identical mtDNA genome sequences (*Figure 1—figure supplement 1b*, *Supplementary file 1*). However, the skull was found in room four of the cave, whereas the femur and metatarsal were found in room three. The available evidence therefore suggests that these samples represent at least two individuals.

Intriguingly, we further note that, across all 17 NWSL equid samples, the relative mapping frequency for chromosomes 8 and 13 is appreciably greater than the remaining autosomes (*Appendix 2—table 4—source data 1*). This may suggest that duplicated regions of these chromosomes are present in NWSL equids, as compared to the horse (*E. caballus*).

## Designation of a type species for *Haringtonhippus*

We sought to designate a type species for the NWSL equid genus, *Haringtonhippus*, using an existing name, in order to avoid adding to the unnecessarily extensive list of Pleistocene North American equid species names (*Winans, 1985*). For this, we scrutinized nine names that have previously been assigned to NWSL equids in order of priority (date the name was first described in the literature). We rejected names that were solely based on dentitions, as these anatomical features are insufficient for delineating between equid groups (*Groves and Willoughby, 1981*). The earliest named species with a valid, diagnostic holotype is *francisci Hay (1915)*. On the basis of taxonomic priority, stratigraphic age, and cranial and metatarsal comparisons (see main results and below), we conclude that *francisci Hay (1915)* is the most appropriate name for *Haringtonhippus*. We note that this middle Pleistocene species is also small, like our late Pleistocene specimens.

The nine examined names were:

*conversidens* Owen, 1869: a small species based upon a partial palate from Tepeyac Mountain, northeast of Mexico City, Mexico. The type fossil has no reliably diagnostic features other than small size, and no more diagnostic topotypal remains are available. For this reason, the validity of the name has previously been challenged by some authors (e.g.,

*Winans, 1985*; *MacFadden, 1992*). However, *Scott, 2004* argued for retaining the name because of its long history of use and utility in promoting taxonomic stability; that study explicitly considered the species to be a small, stout-limbed equid, following the conventions of numerous previous investigations. Following this interpretation, the name *conversidens* would not be available for NWSL equids assigned herein to *Haringtonhippus*. We note in this context that *Barrón-Ortiz et al. (2017)* obtained mtDNA from an equid tooth (EQ30) from San Josecito Cave, Mexico, whose fossil equid assemblage has been assigned by earlier authors (e.g., *Azzaroli, 1992*; *Scott, 2004*) to *Equus conversidens*. Although this fossil assemblage consists of non-NWSL equids, the mtDNA obtained from the tooth indicated placement within the NWSL equid clade (see also *Appendix 2—table 3*). This finding led *Barrón-Ortiz et al. (2017)* to infer some degree of plasticity in the metapodial proportions of the NWSL equids, and to select *conversidens* as their preferred species name for them. We do not follow this interpretation for two reasons: (1) the holotype of the species *conversidens* is nondiagnostic; and (2) selecting a stout-limbed equid species for NWSL equids is problematic.

 *tau* Owen, 1869: a small species erected based upon an upper cheek tooth series lacking the $P^2$ from the Valley of Mexico. Other than small size, the species has no reliably diagnostic features. The holotype specimen has been lost, and no topotypal material is available, and so determining whether or not the species represents a NWSL equid is impossible. *Eisenmann et al., 2008* proposed a neotype specimen for the species, consisting of a cranium (FC 673), but this is rejected here on technical grounds: (1) the proposed neotype fossil was listed as being part of a private collection, which negates its use as a neotype; (2) ICZN rules require that a neotype be 'consistent with what is known of the former name-bearing type from the original description and from other sources' and derive from 'as nearly as practicable from the original type locality . . . and, where relevant, from the same geological horizon or host species as the original name-bearing type'.

 *semiplicatus* Cope, 1893: based upon an isolated upper molar tooth from Rock Creek, Texas. The specimen has been interpreted to be derived from the same species as the holotype metatarsal of '*E*'. *calobatus* Troxell (see below) (*Azzaroli, 1995*; *Quinn, 1957*; *Sandom et al., 2014*).

 *littoralis* Hay, 1913: based upon an upper cheek tooth from Peace Creek, Florida. The tooth is small, but offers no diagnostic features.

 *francisci* Hay, 1915: Named in April of 1915 based upon a partial skeleton, including the skull, mandible, and a broken MTIII (TMM 34–2518). Confidently determined to be a NWSL equid based upon reconstruction of the right MTIII by *Lundelius and Stevens, 1970*.

 *calobatus* Troxell, 1915: Named in June of 1915 based upon limb bones. No holotype designated, but lectotype erected by *Hibbard, 1953* (YPM 13470, right MTIII).

 *altidens* Quinn, 1957: based upon a partial skeleton from Blanco Creek, Texas that exhibits elongate metapodials. Synonymized with *francisci* Hay by *Winans, 1985*.

 *zoyatalis* Mooser, 1958: based upon a partial mandible including the symphyseal region and the right dentary with p2-m3. Synonymized with *francisci* Hay by *Winans, 1985*.

 *quinni* Slaughter *et al*. 1962: based upon a MTIII (SMP 60578) and other referred elements from Texas. Synonymized with *francisci* Hay by *Lundelius and Stevens, 1970* and *Winans, 1985*.

## Anatomical comparison of the *francisci* holotype and Gypsum Cave crania

We compared the holotype of *francisci* Hay (TMM 34–2518) from Texas to the Gypsum Cave cranium (LACM(CIT) 109/156450) from Nevada, the latter of which was assigned to *Haringtonhippus* using palaeogenomic data (*Figure 2—figure supplement 1*). Although there are minor anatomical differences between the two crania, which are outlined below, we consider these to fall within the range of intraspecific variation.

 The skull from Gypsum Cave (GCS) can be distinguished from that of the *francisci* holotype (*f*HS) by its slightly larger size, and markedly longer and more slender rostrum, both

absolutely and as a percentage of the skull length. The rostrum of the GCS is also absolutely narrower; the *f*HS, despite being the smaller skull, is transversely broader at the i/3. The palatine foramina are positioned medial to the middle of the $M^2$ in the GCS, whereas they are medial to the $M^2$-$M^3$ junction in the *f*HS. Viewed laterally, the orbits of the GCS have more pronounced supraorbital ridges than those of the *f*HS. The latter skull also exhibits somewhat stronger basicranial flexion than the GCS. Dentally, the GCS exhibits arcuate protocones, with strong anterior heels and marked lingual troughs in $P^3$-$M^3$; the *f*HS has smaller, triangular protocones with less pronounced anterior heels and no lingual trough or groove. These characters are not thought to result from different ontogenetic stages, since both specimens appear to be of young adults (all teeth in wear and tall in the jaw). Both the GCS and the *f*HS have relatively simple enamel patterns on the cheek teeth, with few evident plications. Not only are the observed differences between these two specimens unlikely to result from ontogeny, they also don't result from sex, since both skulls appear to be females given the absence of canine teeth. The inference of the GCS being female is further supported by palaeogenomic data (*Appendix 2—table 4*).

## Attempt to recover DNA from the *francisci* holotype

We attempted to retrieve endogenous mitochondrial and nuclear DNA from the holotype of *francisci* Hay (TMM 34–2518), to directly link this anatomically-derived species name with our palaeogenomically-derived genus name *Haringtonhippus*, but were unsuccessful.

After sequencing a library enriched for equid mitochondrial DNA (see Appendix 1), we could only align 11 reads to the horse reference mitochondrial genome sequence with BWA. Using the basic local alignment search tool (BLASTn), we show that these reads are 100% match to human and therefore likely originate from contamination. We repeated this approach using MIA and aligned 166 reads, which were concentrated in 20 regions of the mitochondrial genome. We identified these sequences as human (n = 18, 96–100% identity), cow (n = 1, 100%), or Aves (n = 1, 100%), consistent with the absence of endogenous mitochondrial DNA in this sample.

We further generated ~800,000 reads from the unenriched library for TMM 34–2518, and followed a modified metagenomic approach, outlined in (*Graham et al., 2016*), to assess if any endogenous DNA was present. We mapped the reads to the horse reference genome (EquCab2), using the BWA-aln settings of (*Graham et al., 2016*), of which 538 reads aligned. We then compared these aligned reads to the BLASTn database. None of the reads uniquely hit Equidae or had a higher score to Equidae than non-Equidae, whereas 492 of the reads either uniquely hit non-Equidae or had a higher score to non-Equidae than Equidae. These results are consistent with either a complete lack, or an ultra-low occurrence, of endogenous DNA in TMM 34–2518.

## Morphometric analysis of third metatarsals

Stilt- and stout-legged equids can be distinguished with high accuracy (98.2%; logistic regression) on the basis of third metatarsal (MTIII) morphology (*Figure 2c*, *Appendix 1— table 2—source data 1*, and *Appendix 2—table 4—source data 1*), which has the potential to easily and confidently distinguish candidates from either group prior to more costly genetic testing. We note that future genetic analysis of ambiguous specimens, that cross the 'middle ground' between stilt- and stout-legged regions of morphospace, could open the possibility of a simple length-vs-width definition for these two morphotypes. Furthermore, we can highlight potential misidentifications, such as the two putative *E. lambei* specimens that fall within stilt-legged morphospace (*Figure 2c*), which could then be tested by genetic analysis. Intriguingly, an Old World *E. ovodovi* (stilt-legged; MT no. 6; [*Eisenmann and Sergej, 2011*]) and New World *E.* cf. *scotti* (stout-legged; CMN 29867) specimen directly overlap in a stout-legged region of morphospace (*Figure 2c*), which could indicate that either this *E. ovodovi* specimen was misidentified or that this species straddles the delineation between stilt- and stout-legged morphologies.

*H. francisci* occupies a region of morphospace distinct from caballine/stout-legged *Equus*, but overlaps considerably with hemionine/stilt-legged *Equus* (**Figure 2c**). The holotype of *H. francisci* (TMM 34–2518) is very pronounced in its slenderness; it has a greater MTIII length than most other *H. francisci* but slightly smaller width/breadth measurements. This holotype is surpassed in these dimensions only by the *quinni* Slaughter *et al*. holotype, which has itself previously been synonymized with *francisci* Hay (**Lundelius and Stevens, 1970**; **Winans, 1985**). This suggests a potentially larger range of MTIII morphology for *H. francisci* than exhibited by the presently assigned specimens. We observe that this diversity may be influenced by geography, with *H. francisci* specimens from high-latitude Beringia having shorter MTIIIs relative to those from the lower-latitude contiguous USA.

We note that two New World caballine *Equus* from Yukon, *E*. cf. *scotti* and *E. lambei*, appear to separate in morphospace (**Figure 2c**), primarily by MTIII length, supporting the potential delineation of these two taxa using MTIII morphology alone.

