## [Decision Letter]

Thank you for submitting your article "A new genus of horse from Pleistocene North America" for consideration by *eLife*. Your article has been reviewed by four peer reviewers, and the evaluation has been overseen by a Reviewing Editor and Diethard Tautz as the Senior Editor. The following individuals involved in review of your submission have agreed to reveal their identity: Jose Luis Prado (Reviewer #1); Anna Linderholm (Reviewer #2); Maria Teresa Alberdi (Reviewer #4).

The reviewers have discussed the reviews with one another and the Reviewing Editor has drafted this decision to help you prepare a revised submission.

Summary:

This study uses paleogenomic data (from both the mitochondrial and nuclear genome) to convincingly revise the taxonomic diversity and position of the extinct New World stilt-legged (NWSL) equids. Based on their results, the authors propose a new genus, *Haringtonhippus*, for what they demonstrate is a single NWSL species, *H. francisci.*

Reviewers praised the thoroughness of the ancient DNA and phylogenomic analysis; the taxonomic revision itself is convincing. There are two essential revisions that the authors must address before the paper can be considered further for publication in *eLife*.

Essential revisions:

1) There were questions about whether the authors are justified to name a new genus based on the results as presented; demonstration that the NWSL are a sister taxon to non-NWSL *Equus* spp. is insufficient on its own. Of course there is not a straightforward cutoff available, but can the authors provide a comparative sense of scale based on the range of phylogenetic divergence (e.g. nucleotide divergence) for other genus-genus pairs in the same order? Showing comparability to conventional genus pairs in the order would help to justify the decision by the authors to name a new genus; otherwise the conservative choice may be to not do so.

2) Somewhat related to the above point, the estimated divergence time for the NWSL is much older than the actual fossil record seems to suggest. This could easily be a result of poor fossil records or it could be that the genetic analysis is not foolproof (several other studies has shown that using molecular clocks as part of divergence estimates can be problematic). The authors have some treatment of this discordance in the Discussion, but they should consider more fully the potential sources of inaccuracy in the genomic-based divergence estimates.

---

## [Author Response]

Essential revisions:1) There were questions about whether the authors are justified to name a new genus based on the results as presented; demonstration that the NWSL are a sister taxon to non-NWSL Equus spp. is insufficient on its own. Of course there is not a straightforward cutoff available, but can the authors provide a comparative sense of scale based on the range of phylogenetic divergence (e.g. nucleotide divergence) for other genus-genus pairs in the same order? Showing comparability to conventional genus pairs in the order would help to justify the decision by the authors to name a new genus; otherwise the conservative choice may be to not do so.

We appreciate that the rules governing taxonomy are somewhat fluid, and also that our decision to create a new genus may be seen as not particularly conservative. We argue, however, that our decision is robust given recognized taxonomy. Contrary to all previous phylogenetic work on the NWSL equids, which suggests that they fall within genus *Equus*, our work establishes that it does not. To be so included, the last common ancestor of all recognized species of *Equus* would have to also be the ancestor of NWSL equids. Our palaeontological and phylogenetic analyses strongly indicate that any such ancestor would likely also have been ancestral to many other taxa not normally included in *Equus*, resulting in undesirable paraphyly. According to nomenclatural rules, because the species-level taxon comprising NWSL equids falls outside *Equus* as narrowly defined, it must be included in another, cognate clade at the genus level. In the absence of a pre-existing, valid name for this newly-recognized clade, we have fashioned our own, *Haringtonhippus.* To clarify this in the manuscript, we have refined the first paragraph of the systematic palaeontology section.

In addition, and to address this suggestion directly, we investigated during the course of this work the use of phylogenetic/nucleotide divergence to set a genus-genus cutoff. Unfortunately, the distributions of within-genus (species-species) and within-family (genus-genus) nucleotide divergences of mammals are strongly overlapping, including the within-*Equus* nucleotide divergences (Johns and Avise 1998). Within the order Perissodactyla and based on our full/reduced mitochondrial genome alignments (dataset four), species in the tapir genus (*Tapirus*) have a greater mean nucleotide divergence (10.6%/4.9%) than between-genus rhinoceros pairs (e.g. *Ceratotherium-Diceros:* 7.1%/3.3%; and *Dicerorhinus-Coelodonta:* 8.9%/3.5%). This suggests that mean nucleotide divergence is not consistent with taxonomic rank in our study group, and so would not be a conservative approach to discriminate genera here.

Johns, G. C. & Avise, J. C. (1998) A comparative summary of genetic distances in the vertebrates from the mitochondrial cytochrome b gene. *Molecular Biology and Evolution, 15*(11), 1481961490.

2) Somewhat related to the above point, the estimated divergence time for the NWSL is much older than the actual fossil record seems to suggest. This could easily be a result of poor fossil records or it could be that the genetic analysis is not foolproof (several other studies has shown that using molecular clocks as part of divergence estimates can be problematic). The authors have some treatment of this discordance in the Discussion, but they should consider more fully the potential sources of inaccuracy in the genomic-based divergence estimates.

This observation is correct - the fossil record suggests that NWSL equids appear ~2-3 Ma and caballine horses ~1.9-0.7 Ma, and our molecular clock-based estimates suggest that these lineages are at least twice as old as this. Differences between fossil and genetic divergence estimates are not uncommon, and this can be due to a variety of issues including an incomplete fossil record or discordance between the appearance of morphologically distinctive traits and genetically distinct lineages, poor or insufficient calibration of the molecular clock or variations in the rate of molecular evolution over time, errors in phylogenetic estimates, and a combination of these. In this specific case, it seems unlikely that the incompleteness of the fossil record could explain this discrepancy, as the equid fossil record is well described in North America. However, our molecular clock-based estimates are based on a previously estimated divergence between caballine horses and donkeys of 4.0-4.5 Ma, which is a range that was estimated by Orlando et al., 2013 in their analyses of living and a very old (560-780,000-year-old) horse genome. In fact, the observation that these divergence estimates were approximately twice as old as generally accepted was first explored in this 2013 manuscript, where the authors also noted that their result was in agreement both with previous molecular data (Vilstrup et al., 2013) and with the age of a fossil horse from Mexico, *Dinohippus mexicanus*, which is considered a direct cladogenetic ancestor of early *Equus* (MacFadden and Carranza-Castaneda 2002).

To explore this incongruence more thoroughly in our manuscript, we have revised the text in the Discussion section 'Reconciling the genomic and fossil records of Plio-Pleistocene equid evolution'. We note that this type of problem is not unique to equids, and certainly one into which genomic data from increasingly diverse lineages may help to improve, in particular as methods to recover genomic data from increasingly old and poorly preserved remains improve.

Orlando, L. *et al*. Recalibrating *Equus* evolution using the genome sequence of an early Middle Pleistocene horse. *Nature*, 499(7456), 749678 (2013).

Vilstrup, J. T. *et al.* Mitochondrial phylogenomics of modern and ancient equids. *PLoS ONE* 8, e55950 (2013).

MacFadden, B. J. & Carranza-Castaneda, O. Cranium of *Dinohippus mexicanus* (Mammalia Equidae) from the early Pliocene (latest Hemphillian) of central Mexico and the origin of *Equus. Bulletin of the Florida Museum Natural History* 43, 16396185 (2002).